# MoveGPT: Scaling Mobility Foundation Models with Spatially-Aware Mixture of Experts

## Abstract

The success of foundation models in language has inspired a new wave of general-purpose models for human mobility. However, existing approaches struggle to scale effectively due to two fundamental limitations: a failure to use meaningful basic units to represent movement, and an inability to capture the vast diversity of patterns found in large-scale data. In this work, we develop MoveGPT, a large-scale foundation model specifically architected to overcome these barriers. MoveGPT is built upon two key innovations: (1) a unified location encoder that maps geographically disjoint locations into a shared semantic space, enabling pre-training on a billion scale; and (2) a Spatially-Aware Mixture-of-Experts Transformer that develops specialized experts to efficiently capture diverse mobility patterns. Pre-trained on billion-scale datasets, MoveGPT establishes a new state-of-the-art across a wide range of downstream tasks, achieving performance gains of up to 35% on average. It also demonstrates strong generalization capabilities to unseen cities. Crucially, our work provides empirical evidence of scaling ability in human mobility, validating a clear path toward building increasingly capable foundation models in this domain. The source code and pre-trained models for MoveGPT are publicly available at: `https://anonymous.4open.science/r/MoveGPT-FC72/`.

## 1 Introduction

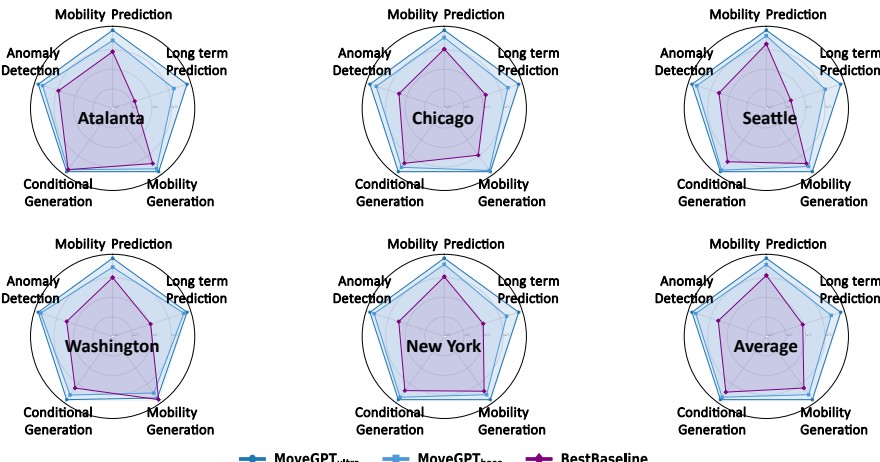

Figure 1: Performance of MoveGPT and the best baseline.

Understanding human mobility patterns is a cornerstone for gaining insight into complex human behaviors and socio-economic dynamics (Song et al., 2010a; Gonzalez et al., 2008). The vast digital traces from billions of personal devices have provided researchers with massive datasets of human mobility, capturing daily activities on an unprecedented scale (Feng et al., 2018; Yuan et al., 2022; Xue et al., 2021). The effort to find predictable patterns within this apparent randomness is not new;

Table 1: Comparison between existing models and MoveGPT across key aspects.

| Method | Model Init. | Unit | Temp. Regularity | Geo. Scalability | Task Flexibility | #Param |
|---|---|---|---|---|---|---|
| TrajFM (Lin et al., 2024) | Scratch | Raw GPS | Irregular | ✓ | ✗ | 18M |
| UniTraj (Zhu et al., 2024) | Scratch | Raw GPS | Irregular | ✓ | ✓ | 2.3M |
| TrajGPT (Hsu et al., 2024) | Scratch | Stay Point | Irregular | ✗ | ✗ | 20w |
| TrajGDM (Chu et al., 2023) | Scratch | Stay Point | Irregular | ✗ | ✗ | 28M |
| UniMove (Han et al., 2025) | Scratch | Stay Point | Irregular | ✓ | ✗ | 40M |
| GenMove (Long et al., 2025a) | Scratch | Zone ID | Regular | ✗ | ✗ | 21M |
| UniMob (Long et al., 2025b) | Scratch | Zone ID | Regular | ✗ | ✓ | 6M |
| UniST (Yuan et al., 2024a) | Scratch | Zone ID | Regular | ✓ | ✓ | 30M |
| UrbanGPT (Li et al., 2024b) | LLMs | Zone ID | Regular | ✓ | ✗ | - |
| UrbanDiT (Yuan et al., 2024b) | Scratch | Zone ID | Regular | ✓ | ✓ | 30M |
| MoveGPT | Scratch | Stay Point | Irregular | ✓ | ✓ | 261M |

nearly two decades ago, studies revealed surprising regularities and high predictability in human mobility (Song et al., 2010b; Schneider et al., 2013). This combination of massive data and intrinsic predictability creates the ideal conditions to move beyond task-specific statistical models and towards a more holistic and generalizable understanding of human movement (Zhou et al., 2024).

Inspired by the transformative success of foundation models in language, a new wave of general-purpose models for mobility has emerged, such as UniST (Yuan et al., 2024a), UrbanDiT (Yuan et al., 2024b), TrajFM (Lin et al., 2024), TrajGPT (Hsu et al., 2024), UniMob (Long et al., 2025b), GenMove (Long et al., 2025a), and UniTraj (Zhu et al., 2024). However, these pioneering efforts, while valuable, fall short of creating a true "GPT for mobility". A detailed comparison, summarized in Table 1, reveals several practical limitations shared by current approaches. Specifically, they rely on inconsistent basic units to represent movement, from raw GPS coordinates to non-transferable Zone IDs, a fragmentation that directly hinders their geographic scalability and prevents cross-city pre-training. Furthermore, many of these models exhibit limited task flexibility and are constrained in their parameter scale, which prevents them from fully leveraging the rich patterns within large, diverse mobility datasets.

We argue these practical limitations stem from two fundamental yet unsolved barriers. The first barrier is the lack of a universal basic unit, which directly prevents the scaling of data across geographies. For instance, some models define tokens as aggregated spatio-temporal patches (e.g., UniST (Yuan et al., 2024a), UrbanDiT (Yuan et al., 2024b)), an approach too coarse to capture the semantics of individual movement. Others use raw GPS coordinates (e.g., UniTraj (Zhu et al., 2024)), which are too fragmented to represent daily routines, or rely on city-specific Zone IDs (e.g., TrajGPT (Hsu et al., 2024), UniMob (Long et al., 2024), GenMove (Long et al., 2025a)), which are not universal and prevent the creation of large-scale, cross-city pre-training datasets. The second barrier is architectural. Human mobility, much like language, is a mixture of distinct behaviors shaped by varied intents and geographical contexts (Han et al., 2025). Existing models, with their "one-fits-all" parameters, are forced to learn an inefficient, averaged representation of these diverse patterns, failing to capture the rich semantics of movement as data diversity grows. Recent LLM-based approaches (Wang et al., 2023; Li et al., 2024a; Feng et al., 2025; Du et al., 2024; 2025) sidestep this by reasoning in a language space, a process that divorces the model from the data's intrinsic spatio-temporal structure.

In this work, we introduce MoveGPT, a scalable and unified architecture for pre-training large-scale foundation models on massive human mobility datasets. To address the representation challenge, MoveGPT introduces a unified location encoder that maps all locations into a universal semantic feature space based on their geographic, functional, and social characteristics. This unlocks the ability to pool large-scale datasets, enabling us to construct a pre-training corpus of unprecedented scale. To tackle the architectural challenge, we design MoveGPT as a decoder-only Transformer with a Spatially-Aware Mixture-of-Experts (SAMoE) architecture. This allows to develop specialized sub-networks ("experts") for distinct spatial contexts and behavioral patterns, efficiently learning a rich understanding of movement as data scales. Operating in an auto-regressive manner, this framework is inherently flexible, supporting various tasks and prediction horizons. Pre-trained on a massive dataset of over 1.5 billion samples from more than 16 cities, MoveGPT establishes a new state-of-the-art on a wide range of downstream tasks and demonstrates remarkable generalization ability to new cities. Further analysis confirms the model's ability to create a unified urban feature space while capturing the hierarchical nature of human mobility. Our work provides empirical evidence of scaling ability in human mobility, demonstrating predictable performance gains from data volume, model size,

and data diversity. This validates a clear and principled path toward building increasingly capable foundation models for human mobility. Our key contributions are as follows:

- To our best knowledge, we are the first to scale mobility foundation models by tackling the dual barriers of location heterogeneity across geographies and pattern heterogeneity in behaviors.
- We introduce MoveGPT, a novel architecture featuring a unified location encoder for multi-city representation and spatially-aware mixture-of-experts Transformers to efficiently model diverse movement patterns.
- MoveGPT establishes a new state-of-the-art on a wide range of downstream tasks, with improvements of up to 30% in prediction and 80% in generation. We present empirical evidence of scaling ability in human mobility.

## 2 PRELIMINARY

**Human mobility data.** Human mobility datasets record the movement trajectories of individuals. Unlike raw GPS data which records continuous movement, our work focuses on trajectories composed of semantically meaningful stay points, which are significant locations where an individual spends time. An individual's movement trajectory $S_u$ is thus represented as a time-ordered sequence of these stay points:

$$S_u = \langle \tilde{t}_1, loc_1 \rangle, \langle \tilde{t}_2, loc_2 \rangle, ..., \langle \tilde{t}_n, loc_n \rangle \tag{1}$$

where $\tilde{t}_i$ denotes the timestamp and $loc_i$ is the i-th stay point, forming an irregular time series. For computational representation, each unique location $loc_i$ is mapped to a cell within a non-overlapping grid. Furthermore, we enrich each location with high-level features (e.g., functionality and geographical attributes) to achieve a more expressive representation.

**General mobility modeling.** To comprehensively evaluate our model's capabilities, we address five fundamental tasks in human mobility modeling. These tasks provide complementary perspectives on understanding, simulating, and evaluating real-world movement behaviors: (1) **next-location prediction:** forecasting the next location based on a user's historical trajectory; (2) **long-term prediction:** predicting a user's trajectory over an extended future time horizon; (3) **unconditional generation:** generating realistic mobility trajectories that reflect typical movement patterns within a city; (4) **conditional generation:** generating trajectories that adhere to specific, given constraints or user attributes (e.g., a certain radius of gyration); (5) **anomaly detection:** identifying abnormal trajectories that deviate from established patterns, potentially due to issues like sensor malfunctions. Successfully addressing these diverse tasks is essential for advancing the development of truly general-purpose mobility foundation models.

## 3 METHODOLOGY

Figure 2 illustrates the overall architecture of our proposed MoveGPT. Panel (a) presents the autoregressive pretraining of mobility data, implemented with a decoder-only transformer enhanced by causal attention and a spatially-aware mixture-of-experts (SAMoE). Panel (b) shows the unified encoding of locations across different cities. Panel (c) depicts how feature embeddings of all city locations are processed through a Deep & Cross Net to generate candidate embeddings, which facilitates a scalable location selection.

### 3.1 UNIFIED LOCATION ENCODER

Existing approaches typically rely on discrete IDs to encode locations. However, such representations hinder cross-city transferability. To build a mobility foundation model, it is essential to develop a unified encoder that can generate consistent representations of locations across different cities. Existing research (Hu et al., 2020; Song et al., 2010c; Gonzalez et al., 2008; Lenormand et al., 2020) shows evidence that human mobility is influenced by individual intents, travelers tend to choose their next destination based on travel needs and spatial (Calabrese et al., 2010; Bontorin et al., 2025; Long et al., 2025b). To capture this behavior, we use the locations' functional feature (POI distribution) and geographic feature (geographic coordinates) as core features for the encoder. In addition, mobility patterns are also influenced by broader social dynamics. To account for this, we incorporate the

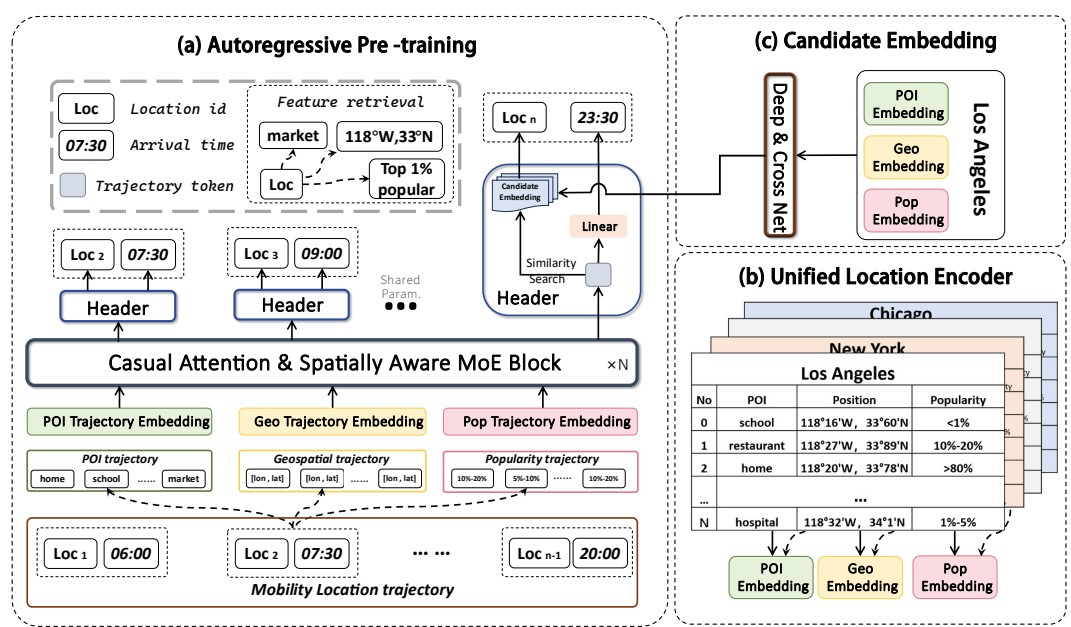

Figure 2: Illustration of the whole framework of MoveGPT, including three key components: (a) Autoregressive pretraining for mobility modeling; (b) Unified location encoder; (c) Candidate embedding for similarity search.

social popularity level of each location as another key feature. Formally, for all locations in a city, we construct three embeddings:

$$\mathbf{E}_{POI} = \text{MLP}(\mathcal{P}), \ \mathbf{E}_{Geo} = \text{MLP}(\mathcal{G}), \ \mathbf{E}_{pop} = \text{MLP}(\mathcal{R}) \tag{2}$$

which $\mathcal{P}, \mathcal{G}$, and $\mathcal{R}$ represent POI distribution, geographic coordinates, and popularity level of the location. This encoding strategy not only preserves the uniqueness of each location but also projects spatial representations from different cities into a shared feature space, thereby enabling more effective and transferable modeling.

## 3.2 SPATIALLY-AWARE MIXTURE-OF-EXPERTS TRANSFORMER

To build mobility foundation models, another key challenge lies in the heterogeneity of individual mobility patterns. To address this, we design spatially-aware mixture-of-experts (SAMoE) Transformers, which leverage causal attention to capture the sequential patterns of mobility trajectories. The key departure from conventional MoE architectures is our spatially-aware routing mechanism, which intelligently assigns trajectory segments to specialized experts based on their spatial context and semantics.

**Semantic-Aware Sequence Input.** The mobility trajectory $S_u = \{loc_1, loc_2, \ldots, loc_n\}$ is mapped into three new sequences through the location feature retrieval method illustrated in Figure 2:

$$S_f = \{x_1, x_2, \ldots, x_n\}, \text{where } x_i = (\text{POI}_i, \text{Geo}_i, \text{Pop}_i). \tag{3}$$

Each sequence is individually embedded to obtain $E_{\text{poi}}$, $E_{\text{geo}}$, and $E_{\text{pop}}$. The fused trajectory embedding is initialized as:

$$E_{\text{traj}} = E_{\text{poi}} + E_{\text{geo}} + E_{\text{pop}}. \tag{4}$$

Additionally, temporal information of the trajectory is embedded into $E_{\text{ts}}$, defined as:

$$E_{\text{ts}} = \text{EMB}(tod) + \text{EMB}(dow) + \text{EMB}(dur), \tag{5}$$

where $\text{Emb}(\cdot)$ denotes an embedding layer, $tod$ is the time of day, $dow$ is the day of week, and $dur$ is the stay duration. $E_{\text{ts}}$ replaces the traditional positional encoding in Transformer architectures. It is combined with trajectory embeddings (*e.g.,* $E_{\text{traj}}$) and serves as input to the attention

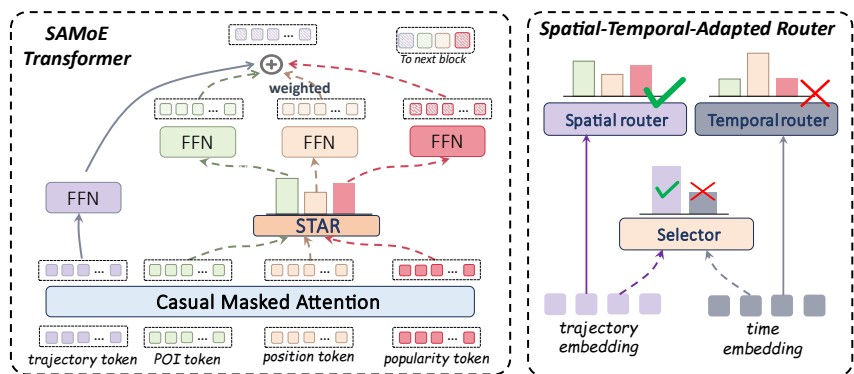

Figure 3: SAMoE Transformer block and SpatiaL-Temporal-Adapted Router(STAR).

layers. As shown in Figure 3, fused trajectory embedding $E_{\mathrm{traj}}$ and three foundation sequence embedding($E_{\mathrm{poi}}$,$E_{\mathrm{geo}}$,$E_{\mathrm{pop}}$) are processed separately by casual attention layer and through residual connections and layer normalization, to obtains the token representations $H$ of the four trajectories as input to the SAMoE block.

$$H_i = \text{CASUAL ATTENTION}(E_i), \text{where } i \in (\text{traj}, \text{poi}, \text{geo, pop}). \tag{6}$$

**Spatially-Aware Mixture-of-Experts.** Human mobility is not driven by a single, monolithic principle. Instead, an individual's movement results from a complex mixture of different intentions and constraints. For example, some movements are driven by personal preference and location function (e.g., choosing a specific restaurant, captured by POI features), while others are primarily sensitive to distance and travel cost (governed by geographic proximity). A third factor is social influence, where individuals are drawn to popular locations (reflected by popularity features). A single, monolithic model would be forced to learn an inefficient, averaged representation of these often-competing drivers. To address this, we design a Spatially-Aware Mixture-of-Experts framework. As illustrated in Figure 3, we instantiate separate experts to specialize in each of these core mobility drivers: a POI expert, a Geo expert, and a Pop expert. These experts operate in parallel, and their outputs are dynamically combined by a Spatial-Temporal-Adapted Router (STAR). STAR acts as a gating mechanism, analyzing the current trajectory context to adaptively weight the influence of each expert for the final prediction, thereby capturing the heterogeneous nature of real-world mobility decisions. STAR's operation can be formally expressed as:

$$g_s^i = \mathbf{W}_s^i(E_{\mathrm{traj}}), \ \ g_t^i = \mathbf{W}_t^i(E_{\mathrm{ts}})$$
$$[w_1, w_2] = \text{Linear}(E_{\mathrm{traj}} \| E_{\mathrm{ts}}), \ \ w = \begin{cases} 1, & \text{if } w_1 \geq w_2 \\ 0, & \text{otherwise} \end{cases} \tag{7}$$
$$g_i = w \cdot g_s^i + (1 - w) \cdot g_t^i$$

where $\mathbf{W}$ is a learnable parameter matrix. $g_s$ and $g_t$ denote the feature importance scores provided by the spatial router and the temporal router, while $w$ serves as an adaptive router selector. The overall formulation of SAMoE block can be expressed as:

$$g_i = \text{STAR}(H_{\mathrm{traj}}, E_{\mathrm{ts}}),$$
$$H'_{\mathrm{traj}} = \text{FFN}_{shared}(H_{\mathrm{traj}}) + \sum_i g_i \text{FFN}_i(H_i). \quad \forall i \in \{\text{poi}, \text{geo}, \text{pop}\} \tag{8}$$

### 3.3 NEXT-STEP OUTPUT

The final output layer is also a major bottleneck in scaling mobility models. Traditional approaches that use a linear projection to produce a softmax distribution over all possible locations are inherently non-scalable, as the output dimension is fixed and cannot adapt to new or changing locations. To overcome this, we design a flexible and scalable pre-training task based on a retrieval framework.

Instead of direct classification, we model next-step prediction as a similarity search problem. Specifically, a mobility trajectory $S$ is encoded by the SAMoE Transformer to produce a trajectory representation $H_{\text{traj}}^l \in \mathbb{R}^d$ for a given length $l$, which acts as a query. Concurrently, we create a candidate database $\mathcal{D} \in \mathbb{R}^{N \times d}$ containing embeddings for all $N$ locations in the city, generated by a Deep & Cross Network (DCN) (Wang et al., 2017) from their features ($E_{\text{poi}}$, $E_{\text{geo}}$, $E_{\text{pop}}$) as follows:

$$\mathcal{D} = \text{DCN}(E_{poi} + E_{geo} + E_{pop}) \tag{9}$$

$loc^n$ is then determined by the dot-product similarity between the trajectory query and the candidate embedding:

$$\hat{loc}^n = H_{\text{traj}}^l \cdot d^n, \ d \in \mathcal{D} \tag{10}$$

This retrieval-based method is highly scalable, as the set of candidate locations $\mathcal{D}$ can be updated (e.g., adding new points of interest) without any changes to the core MoveGPT architecture. The probability distribution for the next timestamp is generated by a linear layer.

### 3.4 MODEL TRAINING

We pretrain MoveGPT in an autoregressive manner on large-scale, multi-city mobility trajectories. The fundamental task is next-step prediction: for any given trajectory, the model learns to predict the next location and its corresponding timestamp. This joint objective enables the model to simultaneously capture the complex spatial patterns and temporal dynamics inherent in human mobility. The overall training objective is therefore defined as the sum of the cross-entropy losses for both the location and timestamp predictions:

$$\mathcal{L}oss = -\sum_{l=1}^{T} \sum_{n=1}^{N} loc_{l+1}^n \log \hat{loc}_{l+1}^n - \sum_{l=1}^{T} \sum_{m=1}^{M} t_{l+1}^m \log \hat{t}_{l+1}^m. \tag{11}$$

## 4 EXPERIMENTS

We conduct extensive experiments to comprehensively evaluate MoveGPT. We pre-train two versions of our model: MoveGPT$_{base}$, trained on one billion samples from six U.S. cities with a 4-layer SAMoE Transformer, and MoveGPT$_{ultra}$, a larger model trained on 1.5 billion samples from 16 U.S. cities with a 12-layer architecture. Further details on datasets andtraining configurations are available in Appendix C and Appendix D.2.

Our evaluation is comprehensive, addressing the model's capabilities from multiple angles. We first assess its cross-task generalization on a diverse range of downstream tasks, including next-location prediction, long-term prediction, conditional/unconditional generation, and anomaly detection. We then investigate the model's scalability, analyzing its performance as we increase data volume, model size, and the number of cities to test for scaling laws. Furthermore, we examine its transfer learning capabilities to see how effectively it transfers learned knowledge to new cities with minimal fine-tuning data. Finally, we conduct an in-depth analysis of the model's internal mechanisms to understand what the model has learned and how it achieves a unified modeling.

Detailed information on the used dataset, baselines, and model setup is provided in the Appendix C and Appendix D. Further analyses, including ablation studies and additional results, can be found in Appendix E.

### 4.1 BASE TASK: NEXT LOCATION PREDICTION

**Setups.** We compare our multi-city pre-trained models (**MoveGPT**$_{base}$ and **MoveGPT**$_{ultra}$) against state-of-the-art (SOTA) baselines across six major cities. To isolate the benefits of multi-city training versus architectural superiority, we also train a single-city version, **MoveGPT (separate)**, for each city. Performance is measured using Accuracy@k ($Acc@k$).

Table 2: Comparison of MoveGPT against state-of-the-art baselines on Acc@1 and Acc@3. The best and second-best results in each column are highlighted.

| City | Atlanta | | Chicago | | Seattle | | Washington | | New York | | Los Angeles | | Average | |
|------|---------|---------|---------|---------|---------|---------|---------|---------|---------|---------|---------|---------|---------|---------|
| Metric | Acc@1 | Acc@3 | Acc@1 | Acc@3 | Acc@1 | Acc@3 | Acc@1 | Acc@3 | Acc@1 | Acc@3 | Acc@1 | Acc@3 | Acc@1 | Acc@3 |
| Markov | 0.133 | 0.300 | 0.123 | 0.264 | 0.125 | 0.249 | 0.123 | 0.326 | 0.109 | 0.296 | 0.102 | 0.235 | 0.119 | 0.278 |
| LSTM | 0.181 | 0.356 | 0.178 | 0.353 | 0.174 | 0.343 | 0.185 | 0.371 | 0.176 | 0.348 | 0.143 | 0.336 | 0.172 | 0.351 |
| Transformer | 0.164 | 0.337 | 0.160 | 0.317 | 0.158 | 0.324 | 0.159 | 0.345 | 0.163 | 0.355 | 0.135 | 0.327 | 0.156 | 0.334 |
| DeepMove | 0.184 | 0.384 | 0.168 | 0.343 | 0.171 | 0.346 | 0.186 | 0.375 | 0.165 | 0.345 | 0.151 | 0.338 | 0.171 | 0.355 |
| GetNext | 0.190 | 0.375 | 0.186 | 0.364 | 0.186 | 0.363 | 0.203 | 0.394 | 0.183 | 0.366 | 0.159 | 0.334 | 0.184 | 0.366 |
| CTLE | 0.206 | 0.403 | 0.183 | 0.361 | 0.194 | 0.371 | 0.192 | 0.378 | 0.163 | 0.349 | 0.150 | 0.336 | 0.181 | 0.366 |
| TrajFM | 0.208 | 0.402 | 0.191 | 0.361 | 0.209 | 0.381 | 0.201 | 0.407 | 0.186 | 0.372 | 0.155 | 0.337 | 0.192 | 0.377 |
| UniMove | 0.217 | 0.361 | 0.209 | 0.360 | 0.235 | 0.393 | 0.232 | 0.378 | 0.214 | 0.338 | 0.171 | 0.259 | 0.213 | 0.348 |
| TrajGPT | 0.214 | 0.400 | 0.203 | 0.361 | 0.226 | 0.390 | 0.225 | 0.386 | 0.192 | 0.366 | 0.179 | 0.320 | 0.206 | 0.370 |
| Unitraj | 0.222 | 0.402 | 0.197 | 0.362 | 0.223 | 0.391 | 0.214 | 0.410 | 0.196 | 0.374 | 0.181 | 0.339 | 0.205 | 0.380 |
| **MoveGPT(ours)** | | | | | | | | | | | | | | |
| *separate* | 0.249 | 0.400 | 0.236 | 0.363 | 0.265 | 0.425 | 0.254 | 0.403 | 0.235 | 0.374 | 0.221 | 0.336 | 0.243 | 0.383 |
| *base* | 0.272 | 0.428 | 0.258 | 0.392 | 0.283 | 0.447 | 0.277 | **0.432** | 0.261 | 0.407 | **0.241** | 0.363 | 0.265 | 0.411 |
| *ultra* | **0.293** | **0.430** | **0.272** | **0.407** | **0.295** | **0.451** | **0.297** | 0.430 | **0.274** | **0.428** | 0.237 | **0.368** | **0.278** | **0.419** |

**Results.** As shown in Table 2, MoveGPT decisively sets a new state-of-the-art. Notably, joint training on multi-city data provides a significant performance boost of nearly 10% over the single-city versions. This confirms that MoveGPT effectively learns transferable, universal mobility patterns while still capturing city-specific heterogeneity. Moreover, even the single-city MoveGPT achieves SOTA or near-SOTA results, demonstrating the inherent power of the SAMoE architecture itself.

## 4.2 Generalization to Downstream Mobility Tasks

To demonstrate its generalization as a foundation model, we evaluate MoveGPT on a diverse set of downstream tasks: long-term prediction, mobility generation, and anomaly detection.

**Long Term Prediction.** This task tests the model's ability to capture long-range dependencies by predicting an entire day's trajectory based on the previous day. While baseline models suffer from rapidly accumulating errors over longer prediction horizons, MoveGPT's performance remains robust. It outperforms all baselines, achieving average gains of 45% on Acc@1 and 18% on Acc@3. This showcases its superior ability to model long-range spatiotemporal patterns, a critical requirement for multi-step forecasting.

**Unconditional Generation.** Here, the goal is to generate realistic, unprompted daily trajectories that capture the natural patterns of a city. We prompted MoveGPT with random starting points and times to autoregressively generate one-day mobility sequences. To measure realism, we assessed the similarity between generated and real trajectories using Jensen-Shannon Divergence (JSD) across three key attribute distributions: single-trip distance, dwell time, and radius of gyration. As shown in Table 3, MoveGPT's generated trajectories are significantly more realistic than those from baselines, yielding an average **65% improvement** in JSD in cities like Atlanta, Chicago, and Seattle. This result demonstrates that MoveGPT has learned a deep and accurate understanding of the underlying mobility data distribution.

**Conditional Generation.** This more challenging task tests the model's ability to generate trajectories that adhere to a specific, user-defined constraint—in this case, a given radius of gyration. The model must reconcile this explicit constraint with realistic mobility patterns. We evaluate performance on two criteria: realism (JSD of the resulting distribution) and adherence (Mean Absolute Error in km between the target and realized radius of gyration). As detailed in Table 8 of Appendix E.3,

Table 3: Performance of unconditional mobility generation, measured by JSD for trip **Dis**tance, stay **Dur**ation, and **Rad**ius of gyration (**lower is better**). The best and second-best results are highlighted.

| City | Atlanta | | | Chicago | | | Seattle | | | Washington | | | New York | | | Average | | |
|---|---|---|---|---|---|---|---|---|---|---|---|---|---|---|---|---|---|---|
| Models | Dis. | Dur. | Rad. | Dis. | Dur. | Rad. | Dis. | Dur. | Rad. | Dis. | Dur. | Rad. | Dis. | Dur. | Rad. | Dis. | Dur. | Rad. |
| TrajGAN | 0.353 | 0.214 | 0.079 | 0.302 | 0.360 | 0.807 | 0.406 | 0.169 | 0.354 | 0.209 | 0.151 | 0.127 | 0.184 | 0.187 | 0.207 | 0.291 | 0.216 | 0.315 |
| DiffLSTM | 0.203 | 0.100 | 0.045 | 0.150 | 0.200 | 0.304 | 0.182 | 0.089 | 0.180 | 0.093 | 0.078 | 0.079 | 0.088 | 0.096 | 0.119 | 0.143 | 0.112 | 0.145 |
| Movesim | 0.221 | 0.119 | 0.054 | 0.163 | 0.225 | 0.320 | 0.201 | 0.092 | 0.202 | 0.101 | 0.080 | 0.083 | 0.096 | 0.097 | 0.115 | 0.156 | 0.122 | 0.155 |
| Difftraj | 0.121 | 0.057 | 0.017 | 0.097 | 0.121 | 0.221 | 0.126 | 0.040 | 0.089 | 0.058 | 0.036 | 0.042 | 0.045 | 0.044 | 0.054 | 0.089 | 0.060 | 0.084 |
| Unitraj | 0.112 | 0.226 | 0.153 | 0.068 | 0.259 | 0.203 | 0.162 | 0.205 | 0.172 | 0.066 | 0.214 | 0.205 | 0.071 | 0.268 | 0.219 | 0.095 | 0.234 | 0.190 |
| **MoveGPT**$_{base}$ | 0.025 | 0.042 | 0.017 | 0.037 | 0.043 | 0.051 | 0.052 | 0.033 | 0.026 | 0.051 | 0.048 | 0.031 | 0.044 | 0.035 | 0.059 | 0.041 | 0.040 | 0.037 |
| **MoveGPT**$_{ultra}$ | 0.021 | 0.034 | 0.016 | 0.029 | 0.037 | 0.031 | 0.045 | 0.021 | 0.026 | 0.044 | 0.039 | 0.028 | 0.037 | 0.022 | 0.056 | 0.029 | 0.025 | 0.026 |

MoveGPT demonstrates remarkable control. On average, it reduces the JSD by **40%** and the MAE by **27%** compared to baselines. This confirms MoveGPT's advanced capability to generate not just realistic but also steerable and controllable mobility patterns.

**Anomalous Detection.** Finally, we test if MoveGPT's understanding of normal patterns can be leveraged for anomaly detection. By using its learned representations to identify trajectories that deviate from the norm, MoveGPT achieves significant performance improvements over baselines (Appendix Table 9), with gains of approximately **17% in precision and 8% in recall**. This demonstrates that the pre-trained model has implicitly learned a powerful and robust representation of normal human mobility. The fine-tuning settings are shown in Appendix D.3.

### 4.3 SCALABILITY ANALYSIS

We conduct a comprehensive analysis of the scalability of our model from three perspectives: (1) **Data Volume**: increasing the amount of training data while keeping the number of training cities fixed; (2) **Model Parameter**: varying the model parameters under the same training data; (3) **Data Diversity**: starting from a single-city dataset and progressively incorporating additional cities to enrich the diversity of individual mobility patterns. As illustrated in Appendix Figure 9, larger model capacity and a larger amount of training data consistently lead to improved performance, which is consistent with the general understanding of foundation models. Moreover, MoveGPT effectively handles the heterogeneity of mobility data across cities, and incorporating more diverse mobility trajectories during training yields substantial performance gains.

### 4.4 TRANSFER LEARNING

One significant advantage of employing large-scale data for model pretraining lies in the effective transfer of knowledge acquired from pretraining datasets to target cities. This enables the model to achieve superior prediction performance through minimal data fine-tuning in target domains. To systematically validate the model's generalization capability, we conducted experiments with 1%, 5%, and 10% of the target city data for single-epoch fine-tuning on the pretrained model. As illustrated in Figure 4, the results demonstrate that merely **5%** of data for fine-tuning can **outperform** the performance achieved by training non-pretrained models with **full** datasets.

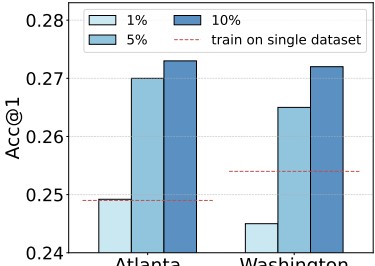

Figure 4: Transfer performance on new cities.

This compelling evidence substantiates the exceptional generalization capacity of our pretraining methodology, highlighting its effectiveness in cross-city knowledge transfer with limited target data requirements.

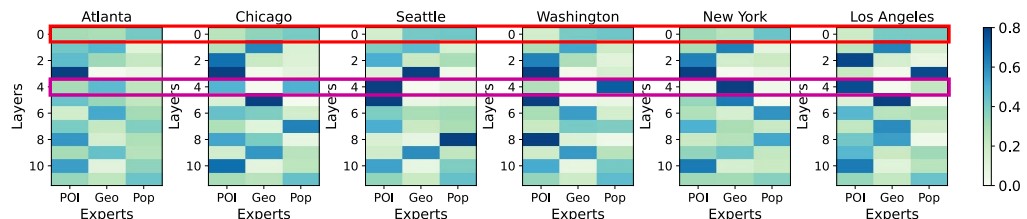

Figure 5: The distribution of the spatial router's weights across different layers in MoveGPT shows both cross-city similarities (e.g., the first layer highlighted in red) and city-specific characteristics (e.g., the fifth layer highlighted in purple).

## 4.5 IN-DEPTH MODEL ANALYSIS

To understand the sources of MoveGPT's strong performance, we conduct an in-depth analysis of its key components. We aim to "open the black box" to interpret what the model has learned and how it handles the complexities of multi-city mobility data.

### 4.5.1 INTERPRETING THE SPATIOTEMPORAL ROUTING MECHANISM

We first analyze the behavior of the Spatial-Temporal-Adapted Router (STAR) to understand how it processes mobility patterns.

**Spatial Router Analysis.** As shown in Figure 5, the spatial router's weights reveal a distinct hierarchical learning pattern across its layers. We observe that the initial and final layers focus on capturing a mixture of fundamental features (geography, POI, popularity). In contrast, the intermediate layers specialize in more abstract semantic patterns, such as complex sequential dependencies and contextual transitions. This layered specialization indicates that MoveGPT effectively integrates low-level spatial data with high-level semantic information to build robust mobility representations.

**Temporal Router Analysis.** The temporal router, analyzed in Appendix Figure 8, shows a predominant and consistent focus on POI semantics across most times of the day and in different cities. This suggests that the model learns that the functional purpose of a location (e.g., dining, working), represented by POIs, is a more stable and powerful indicator of human temporal patterns than mere geographic coordinates.

### 4.5.2 ALIGNMENT OF LOCATION REPRESENTATIONS

A key hypothesis of our work is that a unified location encoder can learn shared spatial semantics across cities. To validate this, we analyze the location embedding layer before and after pretraining. As visualized in Figure 6, the pre-training process transforms the initially scattered, city-specific embeddings into a much more aligned and unified feature space where locations with similar characteristics from different cities cluster together. This provides direct evidence that our encoder successfully bridges the gap between heterogeneous urban environments, which is critical for the model's strong cross-city generalization capabilities.

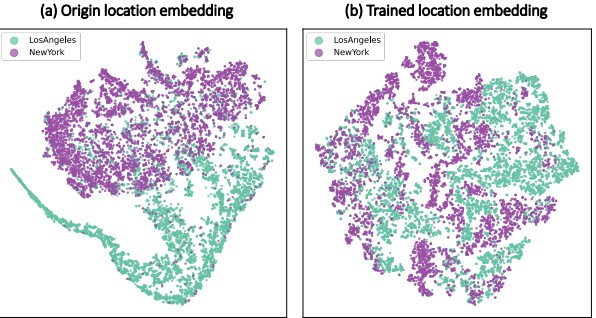

Figure 6: t-SNE visualization of location embeddings for Los Angeles and New York: (a) original embeddings before training, and (b) final embeddings after training.

## 5  CONCLUSION

In this paper, we introduce MoveGPT, a large-scale foundation model that represents a paradigm shift in human mobility research, moving the field beyond the era of fragmented, task-specific models and ushering in a new age of general-purpose, foundational systems. Crucially, our work provides the first empirical evidence of scaling laws in human mobility. We demonstrate that, analogous to the domain of natural language, performance predictably improves with increases in data volume, model size, and data diversity. The implications of MoveGPT extend far beyond academic benchmarks. By laying a robust and scalable foundation, it paves the way for transformative applications in intelligent urban planning, dynamic traffic management, and proactive public health responses. MoveGPT not only advances the state-of-the-art but also charts a clear course for the future of mobility intelligence, promising a deeper, more actionable understanding of the complex dynamics that shape our world.

## ETHICS STATEMENT

The authors have adhered to the ICLR Code of Ethics in the conduct of this research. Our primary ethical considerations were data privacy and the prevention of bias. To protect the privacy of data subjects, we have implemented several robust measures. The dataset used in this study is fully anonymized and contains no personally identifiable information (PII). To further enhance privacy, random noise is added to all location data points (a technique known as location perturbation), making the re-identification of individuals infeasible. Furthermore, the dataset is stored on a secure, encrypted server with access strictly limited to authorized research personnel. Our dataset does not contain any demographic or user-specific attributes, such as gender, race, or age. It inherently mitigates the risk of our model learning and perpetuating societal biases related to these protected characteristics.

Beyond mitigating potential risks, we believe this research has significant positive societal implications. As a foundational tool, MoveGPT is poised to help address major challenges in intelligent urban planning, transportation systems, and public health. We are committed to fostering our research to ensure it contributes constructively to these vital domains.

## REPRODUCIBILITY STATEMENT

In line with our commitment to open and reproducible science, all artifacts required to validate our findings are provided. This includes the complete source code for model training and evaluation, pre-trained model weights, and all configuration files. These materials are available in an anonymized public repository `https://anonymous.4open.science/r/MoveGPT-FC72/` to allow for the verification and extension of our work by the community.

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

## A  USAGE OF LARGE LANGUAGE MODEL

We employed the Google Gemini large language model as an auxiliary writing tool for minor linguistic refinements. Its use was strictly limited to proofreading and improving the grammatical clarity of the text. The substantive content, including all ideas, methods, analyses, and conclusions, is entirely the original work of the authors.

## B  RELATED WORK

### B.1  HUMAN MOBILITY MODELING

Human mobility modeling encompasses a broad spectrum of tasks, including mobility prediction, generation, and anomaly detection. These tasks jointly provide complementary perspectives for understanding, simulating, and evaluating human movement behaviors (Song et al., 2010b; Schneider et al., 2013; Fang et al., 2025). Early approaches were dominated by probabilistic models such as Markov chains and exploration-preferential return (EPR) models (Gambs et al., 2012; Jiang et al., 2016), which captured mobility patterns from transition probabilities but struggled with the irregular and dynamic nature of human behavior. Recent advances in deep learning have significantly boosted the field, with recurrent neural networks (RNNs) (Chen et al., 2020; Yuan et al., 2022), attention-based architectures (Feng et al., 2018; Yuan et al., 2023), Transformers (Corrias et al., 2023; Si et al., 2023), graph neural networks (GNNs) (Sun et al., 2021; Terroso-Sáenz & Muñoz, 2022), and diffusion models (Zhu et al., 2023; Long et al., 2025a) being widely applied across different tasks. However, most of these models remain city-specific, requiring separate training for each dataset and thus limiting their generalizability. Inspired by the paradigm shift in natural language processing, a series of mobility foundation models have recently been proposed, including UniST (Yuan et al., 2024a), UrbanDiT (Yuan et al., 2024b), TrajFM (Lin et al., 2024), TrajGPT (Hsu et al., 2024), UniMob (Long et al., 2025b), GenMove (Long et al., 2025a), and UniTraj (Zhu et al., 2024). These approaches attempt to leverage large-scale pre-training to improve few-shot and zero-shot performance, yet most still can not capture the universality and diversity of human mobility patterns across urban environments.

### B.2  MIXTURE OF EXPERTS

Mixture of Experts (MoE)  (Shazeer et al., 2017; Jordan & Jacobs, 1994; Jacobs et al., 1991) is a deep learning architecture that enhances model performance through a divide-and-conquer strategy. Its core idea lies in decomposing complex tasks and enabling collaborative processing by multiple specialized sub-models (experts), guided by a learnable gating network that dynamically allocates input data to the most relevant experts (Zhou et al., 2022). Each expert processes only a subset of inputs, significantly reducing computational costs, while the gating mechanism synthesizes final results through weighted aggregation of expert outputs (Fedus et al., 2022; Lepikhin et al.).  By maintaining the model's parameter scale while improving inference efficiency, MoE has been widely adopted in large-scale language models (Liu et al., 2024; Du et al., 2022; Lepikhin et al.), enabling trillion-parameter model deployment via sparse activation. However, challenges such as expert load balancing and training stability persist in its implementation.

## C  DATASET DETAILS

We provide detailed information about the dataset used in this study, as shown in the Table 4. The dataset consists of approximately 1.5 billion trajectory points from 16 U.S. cities, each with different population distributions, economic conditions, and urban structures. For pre-processing the trajectory data in these datasets, the trajectory locations of each user are mapped to 500m × 500m grid cells. We applied a sliding window of three days to each user's trajectory and filtered out trajectories with fewer than five trajectory points. For every location, POI data is sourced from Open Street Map and consists of 34 major categories ,and the details are shown in Table 5. The longitude and latitude of all locations in the same city are normalized to have a mean of 0 and a standard deviation of 1. We discretize the popularity rank based on quintiles, and details are shown in Tables 6. For temporal

pre-processing, we divide a day into 48 time slots at half-hour intervals and map $t_i$ to the nearest discrete time point.

Table 4: Basic statistics of mobility data.

| City | Trajectory Point | Temporal Period | User Number | Grid Number |
|---|---|---|---|---|
| Atlanta | 4008w | 2016/10/1-2017/9/30 | 19w | 1175 |
| Chicago | 17361w | 2016/10/1-2017/9/30 | 33w | 1046 |
| Seattle | 6184w | 2016/10/1-2017/9/30 | 9.5w | 3597 |
| Washington | 21415w | 2016/10/1-2017/9/30 | 30w | 1361 |
| Los Angeles | 35616w | 2016/10/1-2017/9/30 | 65w | 4988 |
| New York | 30653w | 2016/10/1-2017/9/30 | 94w | 6198 |
| Boston | 1856w | 2016/3/1-2016/3/14 | 6.2w | 1727 |
| Dallas | 8000w | 2016/3/1-2016/3/14 | 28w | 7936 |
| Miami | 1968w | 2016/3/1-2016/3/14 | 7w | 924 |
| San Francisco | 1744w | 2016/3/1-2016/3/14 | 5.3w | 786 |
| Denver | 3328w | 2016/3/1-2016/3/14 | 10.6w | 5176 |
| Houston | 10880w | 2016/3/1-2016/3/14 | 42.4w | 16550 |
| Minneapolis | 1568w | 2016/3/1-2016/3/14 | 6w | 1368 |
| Philadelphia | 3168w | 2016/3/1-2016/3/14 | 12w | 3525 |
| Phoenix | 4912w | 2016/3/1-2016/3/14 | 18w | 9475 |
| Portland (OR) | 2256w | 2016/3/1-2016/3/14 | 7w | 2804 |

Table 5: POI categories

| | | |
|---|---|---|
| finance | café | dormitory |
| transport | ice cream | shop |
| health | restaurant | travel agency |
| education | boutique | office |
| religion | retail | public transport |
| food | marketplace | home improvement |
| fast food | residential | entertainment |
| shop beauty | recycling | shop transport |
| commodity | sport | pub |
| public | livelihood | service |
| government | accommodation | tourism |
| kindergarten | | |

Table 6: Popularity rank and value

| Popularity Rank | R value |
|---|---|
| <1% | 0 |
| 1%-5% | 1 |
| 5%-10% | 2 |
| 10%-20% | 3 |
| 20%-40% | 4 |
| 40%-60% | 5 |
| 60%-80% | 6 |
| >80% | 7 |

## D  EXPERIMENTS DETAILS

### D.1  BASELINES

- **Markov** (Gambs et al., 2012): The Markov model provides a statistical framework for characterizing how states evolve over time. It predicts future locations by estimating the transition probabilities between states.

- **LSTM** (Graves, 2012): Long Short-Term Memory networks are highly effective for sequential modeling, as they capture long-term dependencies and temporal dynamics, making them particularly suitable for location prediction tasks.

- **Transformer** (Vaswani, 2017): The Transformer architecture leverages self-attention mechanisms to process sequential data efficiently, enabling the modeling of long-range dependencies in human mobility patterns.

- **DeepMove** (Feng et al., 2018): DeepMove is designed to address the challenges of human mobility prediction by jointly considering spatial, temporal, and personal preference factors that shape individual movement behaviors.

- **GetNext** (Yang et al., 2022): GETNext integrates Graph Convolution Networks with Transformers to model global POI transition patterns, incorporating time-aware semantic embeddings for next-location recommendation.

- **CTLE** (Lin et al., 2021): The Context- and Time-aware Location Embedding model dynamically generates location embeddings by combining bidirectional Transformers with temporal encoding, capturing multi-faceted semantics of locations under varying spatial-temporal dynamics.

- **Unimove** (Han et al., 2025): Unimove is a unified model for multi-city human mobility prediction, employing a trajectory-location dual-tower architecture and MoE Transformer blocks to enhance prediction accuracy across diverse urban environments.

- **TrajGPT** (Hsu et al., 2024): TrajGPT is a multitask transformer-based spatiotemporal generative model that frames controlled synthetic trajectory generation as a text infilling problem, ensuring spatiotemporal consistency in visit sequences.

- **TrajFM** (Lin et al., 2024): TrajFM is a trajectory foundation model that emphasizes both region transferability and task transferability through its STRFormer backbone.

- **Unitraj** (Zhu et al., 2024): UniTraj is a universal trajectory foundation model trained on large-scale, worldwide human mobility data, designed to be both task-adaptive and region-independent.

- **Difftraj** (Zhu et al., 2023): DiffTraj is a spatiotemporal diffusion probabilistic model designed for GPS trajectory generation. It employs a Trajectory UNet (Traj-UNet) to reverse a noise process, reconstructing high-fidelity trajectories that preserve the original spatial distributions

- **TrajGAN** (Goodfellow et al., 2014): TrajGAN introduces a GAN framework for trajectory generation, where trajectories perturbed with random noise are mapped into realistic sequences by a generator composed of linear and convolutional layers.

- **DiffLSTM** DiffLSTM is a trajectory generation model that integrates the UNet architecture and ResNet blocks from DiffTraj, replacing convolutional layers with Long Short-Term Memory (LSTM) units in the ResNet blocks. This modification allows the model to capture long-range temporal dependencies in sequential data.

- **Movesim** (Feng et al., 2020): MoveSim is a generative adversarial network (GAN)-based framework for simulating human mobility patterns. It combines model-based and model-free approaches, utilizing SeqNet for sequential transition modeling and RegNet for incorporating urban structure effects, outperforming existing models in metrics like travel distance and location visits.

- **SAE** (Malhotra et al., 2016): SAE is a standard RNN-based sequence-to-sequence architecture trained by minimizing the reconstruction error. The anomaly score for a sequence is determined according to how large its reconstruction error is.

- **GMVSAE** (Liu et al., 2020): GMVSAE is a model designed for detecting anomalies in vessel trajectories by employing a Gaussian Mixture Variational Autoencoder, which analyzes historical data and established shipping lanes to identify vessels that deviate significantly from expected paths.

## D.2 Experiment Configuration

For MoveGPT$_{base}$, we use a 4-layer Transformer, with a hidden dimension of 512 and 4 attention heads. The pretraining data consists of 6 U.S. cities with a total of 6 million trajectories. For MoveGPT$_{ultra}$, we employ a 12-layer Transformer with 6 attention heads and a hidden dimension of 768. The pretraining data includes 16 U.S. cities with a total of 10 million trajectories. All models are trained using the Adam optimizer, with a learning rate of $3 \times 10^{-4}$, a maximum of 50 epochs, and early stopping after 3 epochs without improvement. The batch size is set to 32.

## D.3 Cross Task Finetuning

**Unconditional Generation.** We fine-tune MoveGPT on 1,500 specific city trajectories for the tasks of next location prediction and arrival time prediction. After fine-tuning, the model generates a full day's trajectory by randomly sampling the starting location and time, using a top-k sampling method, and continuing until the predicted time exceeds midnight.

Table 7: Performance of long term prediction(1 day predict 1 day)

| City | Atlanta | | Chicago | | Seattle | | Washington | | New York | | Los Angeles | | **Average** | |
|---|---|---|---|---|---|---|---|---|---|---|---|---|---|---|
| Model | Acc@1 | Acc@3 | Acc@1 | Acc@3 | Acc@1 | Acc@3 | Acc@1 | Acc@3 | Acc@1 | Acc@3 | Acc@1 | Acc@3 | Acc@1 | Acc@3 |
| Transformer | 0.109 | 0.224 | 0.038 | 0.074 | 0.104 | 0.205 | 0.073 | 0.140 | 0.038 | 0.072 | 0.036 | 0.057 | 0.066 | 0.129 |
| DeepMove | 0.112 | 0.243 | 0.045 | 0.107 | 0.115 | 0.239 | 0.092 | 0.200 | 0.056 | 0.147 | 0.055 | 0.158 | 0.079 | 0.182 |
| TrajFM | 0.123 | 0.298 | 0.093 | 0.195 | 0.123 | 0.279 | 0.114 | 0.257 | 0.086 | 0.174 | 0.089 | 0.182 | 0.104 | 0.231 |
| UniTraj | 0.131 | 0.310 | 0.096 | 0.201 | 0.134 | 0.298 | 0.121 | 0.269 | 0.093 | 0.217 | 0.095 | 0.192 | 0.112 | 0.247 |
| **MoveGPT$_{base}$** | 0.170 | 0.347 | 0.127 | 0.234 | 0.176 | 0.349 | 0.163 | 0.304 | 0.126 | 0.247 | 0.125 | 0.221 | 0.148 | 0.284 |
| **MoveGPT$_{ultra}$** | **0.183** | **0.364** | **0.142** | **0.245** | **0.195** | **0.359** | **0.167** | **0.310** | **0.143** | **0.254** | **0.130** | **0.223** | **0.160** | **0.292** |

**Conditional Generation.** We use the embedding of radius of gyration through a linear layer, and add it to the trajectory embedding as the input condition. The rest of the process remains consistent with the unconditional generation.

**Long-Term Prediction.** We use a full-day's trajectory as the prompt and provide the arrival time for the future trajectory. The model then directly predicts the location corresponding to the given time on the following day.

**Anomalous Detection.** We take the last token output by MoveGPT and use it through a linear layer to generate the probability output for anomaly detection.

# E ADDITIONAL RESULTS

## E.1 ABLATION STUDY

SAMoE is the key design of MoveGPT, specifically addressing cross-city data heterogeneity through its specialized gating mechanisms. We conducted systematic ablation studies on five critical configurations: removal of (1) the shared expert, (2) the whole mixture-of-experts, (3) the adapted selector, (4) the temporal router, and (5) the spatial router. Experimental results on Atlanta datasets are shown in Appendix Figure 7. The results demonstrate a consistent performance degradation across all ablated variants. This empirical evidence confirms that each component collectively contributes to effective spatio-temporal pattern modeling, establishing SAMoE's architectural elements as essential design choices for urban mobility modeling.

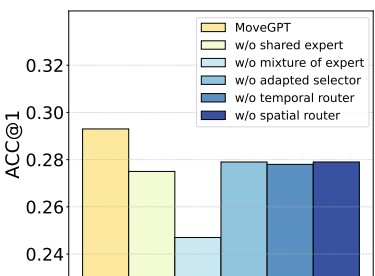

Figure 7: Cross-city transfer performance on the Atlanta and Washington dataset.

## E.2 LONG-TERM PREDICTION

Appendix Table 7 shows the result of Long-term prediction.

## E.3 CONDITIONAL GENERATION

Appendix Table 8 shows the result of conditional mobility generation.

## E.4 ANOMALOUS TRAJECTORY DETECTION

Appendix Table 9 shows the result of anomalous trajectory detection.

## E.5 SPATIAL-TIME-ADAPTED ROUTER ANALYSIS

Appendix Figure 8 shows the top-1 percent of features for the temporal router during a day in MoveGPT.

Table 8: Performance of conditional mobility generation.(Condition: Radius)

| City | Atlanta | | Chicago | | Seattle | | Washington | | New York | | Los Angeles | | **Average** | |
|---|---|---|---|---|---|---|---|---|---|---|---|---|---|---|
| Model | JSD | MAE | JSD | MAE | JSD | MAE | JSD | MAE | JSD | MAE | JSD | MAE | JSD | MAE |
| Diff-LSTM | 0.106 | 2.511 | 0.200 | 3.626 | 0.035 | 1.910 | 0.055 | 1.962 | 0.046 | 5.454 | 0.048 | 4.228 | 0.082 | 3.282 |
| TrajGAN | 0.302 | 7.156 | 0.731 | 8.048 | 0.083 | 5.750 | 0.192 | 4.307 | 0.169 | 11.230 | 0.147 | 10.186 | 0.271 | 7.779 |
| MoveSim | 0.186 | 3.848 | 0.445 | 3.855 | 0.036 | 2.792 | 0.057 | 2.095 | 0.067 | 5.556 | 0.084 | 5.772 | 0.142 | 3.986 |
| DiffTraj | 0.086 | 1.968 | 0.155 | 3.351 | 0.028 | 1.804 | 0.049 | 1.634 | **0.022** | 3.148 | **0.028** | 3.689 | 0.061 | 2.599 |
| Unitraj | 0.067 | 1.803 | 0.069 | 3.274 | 0.042 | 3.642 | 0.045 | 1.667 | 0.036 | 4.287 | 0.086 | 4.40 | 0.057 | 3.185 |
| **MoveGPT$_{base}$** | 0.056 | 1.792 | 0.057 | 2.924 | 0.028 | 1.181 | 0.031 | 1.275 | 0.041 | 2.484 | 0.041 | 3.102 | 0.042 | 2.126 |
| **MoveGPT$_{ultra}$** | **0.049** | **1.627** | **0.042** | **2.548** | **0.025** | **1.074** | **0.029** | **1.036** | 0.033 | 2.247 | 0.030 | **2.866** | **0.034** | **1.899** |

Table 9: Performance of anomalous trajectory detection

| City | Atlanta | | Chicago | | Seattle | | Washington | | New York | | Los Angeles | | **Average** | |
|---|---|---|---|---|---|---|---|---|---|---|---|---|---|---|
| Model | Pre. | Rec. | Pre. | Rec. | Pre. | Rec. | Pre. | Rec. | Pre. | Rec. | Pre. | Rec. | Pre. | Rec. |
| MLP | 0.536 | 0.586 | 0.517 | 0.551 | 0.541 | 0.551 | 0.529 | 0.634 | 0.515 | 0.587 | 0.506 | 0.681 | 0.524 | 0.598 |
| Transformer | 0.651 | 0.629 | 0.697 | 0.677 | 0.705 | 0.675 | 0.677 | 0.623 | 0.668 | 0.745 | 0.718 | 0.776 | 0.686 | 0.687 |
| SAE | 0.659 | 0.600 | 0.701 | 0.655 | 0.707 | 0.651 | 0.643 | 0.601 | 0.630 | 0.743 | 0.690 | 0.748 | 0.672 | 0.666 |
| GMVSAE | 0.763 | 0.675 | 0.743 | 0.745 | 0.749 | 0.704 | 0.715 | 0.650 | 0.716 | 0.822 | 0.753 | 0.833 | 0.740 | 0.738 |
| **MoveGPT$_{base}$** | 0.840 | 0.719 | 0.865 | 0.787 | 0.866 | 0.749 | 0.836 | 0.693 | 0.833 | 0.892 | 0.871 | 0.868 | 0.852 | 0.785 |
| **MoveGPT$_{ultra}$** | **0.861** | **0.733** | **0.899** | **0.818** | **0.891** | **0.760** | **0.848** | **0.699** | **0.853** | **0.927** | **0.879** | **0.881** | **0.871** | **0.803** |

## E.6 ANALYSIS OF SCALABILITY

Appendix Figure 9 shows the result of the scalability performance of MoveGPT.

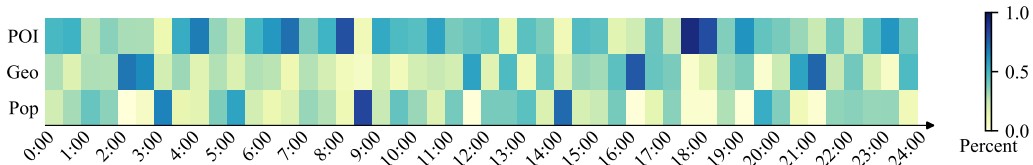

Figure 8: The top-1 percent of temporal router during a day in MoveGPT.

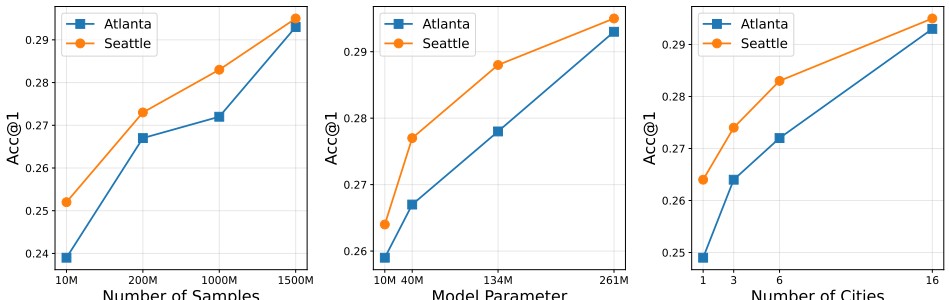

Figure 9: The result of the scalability performance of MoveGPT.

