# OpenReview forum: "MoveGPT: Scaling Mobility Foundation Models with Spatially-Aware Mixture of Experts"
_ICLR.cc/2026/Conference — Submitted to ICLR 2026_

### Official Review · Reviewer_1XkS · 2025-10-25

**Soundness:** 2
**Presentation:** 3
**Contribution:** 2
**Rating:** 4
**Confidence:** 4

**Summary:**

The paper introduces MoveGPT, a large-scale foundation model for human mobility modeling that aims to generalize across multiple cities. The proposed framework combines two key components: (1) a unified location encoder that projects heterogeneous geographic and functional features (e.g., POI distribution, coordinates, popularity) into a shared semantic space; and (2) a Spatially-Aware Mixture-of-Experts (SAMoE) Transformer, which allocates trajectory segments to specialized experts (POI, Geo, Pop) via a spatial–temporal router (STAR). The model is pre-trained on over one billion mobility trajectories from up to 16 cities and evaluated across tasks such as next-location prediction, long-term forecasting, generation, and anomaly detection. Empirical results show strong improvements over prior baselines and good transferability to unseen cities with limited fine-tuning data.

**Strengths:**

1. The paper makes an interesting and timely contribution to the emerging area of mobility foundation models. It presents a large-scale pre-trained model trained on trajectories from 16 cities, demonstrating the feasibility of scaling human mobility modeling to a global level. The dataset size and coverage are impressive and provide a strong empirical foundation for large-scale pretraining.

2. The paper is easy to follow. The motivation, methodology, and experimental setup are logically organized.

3. The paper evaluates across multiple downstream tasks and includes scaling analyses with respect to data volume, model size, and city diversity. The multi-city pretraining and fine-tuning experiments convincingly demonstrate generalization potential.

**Weaknesses:**

1. The design of expert router is questionable. The STAR router is described as a binary selector between spatial and temporal gating, yet it is unclear why a binary mechanism should be used when the model processes three distinct trajectory feature types (POI, coordinates, and popularity). Using either spatial or temporal representations in the MoE process doesn’t make sense to me.

2. Although the model is termed a Spatially-Aware Mixture-of-Experts, the spatial awareness appears limited to the separation of trajectory features (coordinates, popularity, and category). The paper does not clearly explain how this mechanism captures spatial dependencies beyond feature partitioning, making the “spatially-aware” characterization somewhat unconvincing.

3. The paper claims that geographical coordinates are encoded using a shared MLP across all cities. However, since each city has vastly different coordinate ranges and spatial distributions, applying a single encoder directly across cities is questionable. Without normalization strategies that preserve inter-city spatial semantics, it is difficult to believe that this encoding process can be universally applicable.

4. The similarity-based next-location retrieval is essentially the same as token retrieval for generation process in LLMs. However, removing the softmax operation from the output distribution, as implied by Equation (11), raises concerns about whether the resulting scores still represent valid log probabilities.

5. Although the paper claims to release the resources, the referenced datasets do not appear to be publicly available. This restricts the reproducibility of the reported results.

6. The Spatial-aware MoE architecture introduces multiple experts, routers, and feature retrieval modules, which likely increase computational overhead. However, there is no discussion of training time, memory footprint, or inference latency—factors critical for real-world deployment.

**Questions:**

1. The difference between the vector form $\mathbf{g}_i$ and the scalar $g_i$ in Equation (7) is unclear. How are these variables related, and what exactly is the role of each within the MoE computation? The overall formulation of the mixture-of-experts component needs clearer mathematical explanation.

2. Based on Equation 10, how can we make sure that Equation 11 is valid?

3. How is the “next time” (timestamp) predicted in the model? The paper mentions temporal embeddings, but it is not clear how these embeddings are used for time prediction or how they interact with the autoregressive prediction process.

---

> ### Author Response · Authors · 2025-11-16
> **Answer of Weakness and Question**
>
> **W1**: Temporal and spatial information are fundamental attributes of mobility trajectories. Compared with letting the model fully adapt to learning a selector, jointly modeling both aspects helps reduce the learning difficulty and provides additional timestamp information. If what you mean is that a traditional adaptive adapter alone could achieve the effect of MoE training, that is acceptable. However, our method is more effective, and the ablation results demonstrate that each component of STAR is necessary rather than redundant.
>
> **W2**: Geographic location, POI, and popularity features are sufficient for spatial representation. Section 3.1 of the paper describes in detail how these three factors comprehensively model spatial characteristics. We will further extend the model in the future, e.g., adding satellite imagery.
>
> **W3**: The latitude and longitude of all locations within a city are normalized to a normal distribution to represent their relative positions. Different cities cover different spatial ranges, but since the location size is fixed to a 500m grid, the number and distribution of locations implicitly model movement distances between them. User trajectories further support this modeling process.
>
> **W4**: Using vector dot-product as the probability of similarity is not uncommon. In LLM and other fields, cosine similarity and dot-product similarity are widely used in retrieval, so this cannot be regarded as a weakness.
>
> **W5**: The dataset cannot be made public due to user privacy and security concerns. We have updated the anonymous repository with model weights and a small set of example data. We will also release a reasonably validated trajectory dataset generated by the model to support research in this field.
>
> **W6**: We added experiments comparing training time with and without MoE. Overall, since MoE uses only three experts, it does not impose significant computational overhead.
>
> |          | Training time/epoch | Parameters |
> |----------|----------------------|------------|
> | w/ MoE   | 45min 36s              | 261M       |
> | w/o MoE  | 43min 58s              | 152M       |
>
> ---
>
> **Q1**
> We have updated this formula difference in the PDF; the two symbols are equivalent.
>
>  **Q2**
> Equation (11) is the cross-entropy loss for location prediction and time prediction. The probability distribution for location prediction is obtained from Equation (10).
>
>  **Q3**
> Time is divided into 48 half-hour intervals. Time prediction uses the trajectory representation, which is passed through a linear layer to output a probability distribution over the 48 time slots.

---

> ### Comment · Reviewer_1XkS · 2025-11-24
>
> W1: I understand you use two component for spatial and temporal modeling respectively. However, in Eq. (7), $w$ is a **binary** gate. This means that, at each step, the model either uses the spatial part or the temporal part and discards the other. How could it be **joint** modeling as mentioned in your response?
>
> W2: This STAR module is still not clear. In Eq. 7, the input is $E_{traj}$, then in Eq. 8 it becomes $H_{traj}$. What exactly are the shapes of these representations?
>
> W3: The description of coordinate normalization is still too vague. What specific operations did you apply to produce normalized coordinates for cities with different geographic scales and extents? You should give detailed explanation on data processing steps.
>
> W4: If you directly use the dot product as a similarity score feeding into the log-probability calculation in Eq. (11), the dot product can be negative, which is not valid as a probability.
>
> W5: Then your framework is less interesting. I don't think it can act as LLM, which can be easily adaptable as the representation format is natural language, which is universal. The proposed framework is still too specialised to be easily transferable without curated data.
>
> Btw, I have to say that you **did not make a serious effort to digest my comments**. My points in W1 and W4 are completely unaddressed. In addition, for questions like W3, you should provide a detailed explanation rather than saying "we normalized to normal distribution". I don't want to keep repeating the same concerns over and over in my comments.

---

> > ### Author Response · Authors · 2025-11-24
> >
> > **W1: Joint Spatial-Temporal Modeling Clarification**
> >
> > Our joint spatial-temporal modeling is applied **across all users’ trajectories**, rather than on a single trajectory. The binary selector in STAR does **not discard the other component**; it performs adaptive feature selection depending on the trajectory type. Different travel behaviors are driven by different dominant factors: commuting \\( (Home \\rightarrow Work) \\) shows strong temporal regularity, so the temporal router produces more reliable weights; leisure/shopping activities are more sensitive to POI and geographic distance, so the spatial router yields higher-quality gating.
> >
> > Thus, the binary selector in STAR acts as a **trajectory-level dynamic feature selection mechanism**. The “joint modeling” does **not** mean fusing all factors for every trajectory; rather, MoveGPT leverages multiple behavioral drivers **at the global level**, covering diverse travel mechanisms without forcing fusion at the single trajectory level.
> >
> > **W2: Tensor Shapes in STAR and SAMoE**
> >
> > In Eq.(7), \\( E_{ \\text{traj} } \\) should be \\( H_{ \\text{traj} } \\). For clarity, the shapes of all key tensors are:
> >
> > 1. **Input sequences embedding**: batch size \\( B \\), sequence length \\( L \\), hidden dimension \\( d \\)
> > \\[
> > E_{ \\text{traj} },\\; E_{ \\text{poi} },\\; E_{ \\text{geo} },\\; E_{ \\text{pop} } \\in \\mathbb{R}^{B \\times L \\times d}
> > \\]
> >
> > 2. **After attention layers**:
> > \\[
> > H_{ \\text{traj} },\\; H_{ \\text{poi} },\\; H_{ \\text{geo} },\\; H_{ \\text{pop} } \\in \\mathbb{R}^{B \\times L \\times d}
> > \\]
> >
> > 3. **STAR router gating**:
> > Spatial and temporal routers produce weights for three experts (POI/GEO/POP):
> > \\[
> > g_s,\\; g_t \\in \\mathbb{R}^{B \\times L \\times 3}
> > \\]
> > Binary selector \\( w \\in \\{0,1\\} \\) combines them:
> > \\[
> > g = w \\cdot g_s + (1 - w) \\cdot g_t \\in \\mathbb{R}^{B \\times L \\times 3}
> > \\]
> >
> > **W3: Cross-City Coordinate Normalization**
> >
> > We adopt a 3-step process to align coordinates across cities:
> >
> > 1. **Map locations to grid centers**
> >    All location points are first mapped to the centers of **500 m × 500 m grids**.
> >
> > 2. **Compute city-level statistics**
> >    For each city, we collect all valid grid points and calculate the **mean** and **standard deviation** of their longitude and latitude:
> >    \\[
> >    \mu_\text{lon} = \text{mean(lon)}, \quad
> >    \sigma_\text{lon} = \text{std(lon)}, \quad
> >    \mu_\text{lat} = \text{mean(lat)}, \quad
> >    \sigma_\text{lat} = \text{std(lat)}
> >    \\]
> >
> > 3. **Normalization**
> >    Each coordinate is standardized by subtracting the city mean and dividing by the standard deviation:
> >    \\[
> >    lon' = \frac{\text{lon} - \mu_\text{lon}}{\sigma_\text{lon}}, \quad
> >    lat' = \frac{\text{lat} - \mu_\text{lat}}{\sigma_\text{lat}}
> >   \\]
> >
> > This process produces **unified, aligned coordinates** with consistent feature space across cities, improving **cross-city transferability** while preserving relative spatial relationships.
> >
> > **W4: Similarity-Based Retrieval Clarification**
> >
> > Using raw dot-product similarity may produce negative values, which are not valid log probabilities. During training, we use PyTorch’s `F.cross_entropy`, which internally applies log-softmax before computing negative log-likelihood.
> >
> > Thus, the model learns over **log-softmax normalized probabilities**, not raw dot products. This will be clarified in the training section.
> >
> > **W5: Data Availability and Transferability**
> >
> > Full raw trajectory data cannot be shared due to privacy. We provide pretrained model weights, example data, and will release a validated model-generated trajectory dataset for reproducibility.
> >
> > MoveGPT addresses two fundamental challenges in mobility modeling: lack of cross-city unified location representation and high behavioral heterogeneity. Through the Unified Location Encoder and SAMoE, MoveGPT aligns spatial semantics across cities and models diverse mobility drivers via expert decomposition, achieving **scaling and cross-city generalization** similar to language models.
> >
> > **Transfer Experiments**
> >
> > - Using only 5 \% of target city data, MoveGPT matches or exceeds a model trained from scratch on 100 \% data (Fig. 4).
> > - Zero-shot transfer to Minneapolis and Dallas (Acc@1) without fine-tuning:
> >
> > | City        | Separate | Full  | Zero-shot |
> > | ----------- | -------- | ----- | --------- |
> > | Minneapolis | 0.231    | 0.312 | 0.252     |
> > | Dallas      | 0.235    | 0.286 | 0.246     |
> >
> > This demonstrates robust cross-city transfer and learning of universal mobility patterns. All used POI features are publicly available, ensuring reproducibility. Few-shot and zero-shot experiments confirm MoveGPT’s **strong cross-city generalization** and step toward a **general mobility foundation model**.

---

### Official Review · Reviewer_7EEr · 2025-10-29

**Soundness:** 3
**Presentation:** 2
**Contribution:** 2
**Rating:** 2
**Confidence:** 5

**Summary:**

This paper introduces MoveGPT, a large-scale foundation model for human mobility research, aiming to move the field beyond fragmented, task-specific approaches toward general-purpose, foundational systems. The authors propose: 1. A unified location encoder that maps geographically disjoint locations into a shared semantic space, enabling global-scale pretraining; 2. A Spatially-Aware Mixture-of-Experts (MoE) Transformer that leverages specialized experts to efficiently capture diverse mobility patterns. Pretrained on billion-scale datasets, MoveGPT achieves new state-of-the-art results across a wide range of downstream tasks, with average performance improvements of up to 35%.

**Strengths:**

1. The paper introduces the idea of applying Mixture-of-Experts (MoE) to the domain of mobility foundation models, which is relatively novel.

2. The authors conduct extensive experiments, demonstrating SOTA performance on multiple benchmarks.

**Weaknesses:**

1. Overclaiming novelty. The authors repeatedly emphasize their unified location encoder, but similar ideas have already appeared in prior works such as GeoCLIP [1], which also trained a unified location encoder and further extended it to downstream models like UrbanVLP [2]. MoveGPT’s encoder essentially encodes GPS coordinates with auxiliary features such as POIs, which is not a fundamentally new innovation.

2. Manual expert definition in MoE. In large language models, MoE architectures typically allow the model to autonomously learn expert specialization. In contrast, the authors manually define expert categories, which restricts scalability and partially contradicts the spirit of a general foundation model.

3. Incorrect claim about LLM architecture. In line 263, the authors claim that traditional methods use an MLP to predict the output distribution. However, most large models directly employ a linear projection on the final hidden representation, without an additional MLP.

4. Writing and formatting issues. The paper suffers from several writing problems: for example, in line 57, citations are not enclosed in parentheses; in line 200, terms such as POI and GEO are formatted inconsistently with standard academic conventions; and in line 261, there is an empty parenthesis. These suggest that the manuscript was prepared in haste.

5. Code availability mismatch. The abstract claims that both source code and pretrained models are publicly available in the anonymized repository. However, only the code is provided, while no pretrained models are included.

[1] Vivanco Cepeda, Vicente, Gaurav Kumar Nayak, and Mubarak Shah. "Geoclip: Clip-inspired alignment between locations and images for effective worldwide geo-localization." Advances in Neural Information Processing Systems 36 (2023): 8690-8701.

[2] Hao, Xixuan, et al. "Urbanvlp: Multi-granularity vision-language pretraining for urban socioeconomic indicator prediction." Proceedings of the AAAI Conference on Artificial Intelligence. Vol. 39. No. 27. 2025.

**Questions:**

1. Please clarify and respond to the identified weaknesses.

2. Please elaborate on the motivation for adopting a Wide & Deep network. What specific gap does it address in this context? Why is this considered an innovation? Have similar architectures not been explored in previous works on mobility modeling or foundation models?

---

> ### Author Response · Authors · 2025-11-16
> **Answer of Weakness and Question**
>
> First, regarding **Question 2**: we have **never** mentioned a *Wide & Deep* model anywhere in the paper.
> ---
>
> **W1:** Our encoder is specifically designed for *human mobility modeling*. Methods like GeoCLIP and UrbanVIP belong to **different domains**, and their architectures cannot be directly transferred to human mobility tasks. Moreover, our encoder is only **one component** of MoveGPT. MoveGPT is a **human mobility foundation model**, not simply a geo-representation model.
>
> **W2:** Our MoE architecture is **not** the sparse MoE used in LLMs. We manually define three domain experts because these three factors are already sufficient for modeling—this is clearly explained in the paper. If additional features are needed, one can simply add corresponding experts and fine-tune them. Even with a traditional MoE, adding new features would still require retraining, so this is not a scalability issue.
> Furthermore, ablation studies show that each design choice contributes meaningfully and is not redundant.
>
> **W3:** We do **not** use an LLM. The mistake of labeling a linear projection as an “MLP” was indeed our typo. We have corrected this error in the PDF.
>
> **W4:** The formatting issues you pointed out have been corrected.
>
> **W5:** Due to the file size limitations of anonymous repositories, we are currently unable to update the model weights in the repository. We will open-source the model on Hugging Face later. We will further refine the deployment and usage instructions to ensure usability of the model.

---

> > ### Comment · Reviewer_7EEr · 2025-11-17
> >
> > I am sorry for the typo. Actually, my question is focused on the Deep & Cross network.

---

> ### Author Response · Authors · 2025-11-18
> **Answer of Question 2**
>
> We perform ablation studies to validate the effectiveness of the Deep & Cross Network. For fairness, instead of simply removing DCN, we replace it with an MLP.
> The table below reports retrieval accuracy in two cities:
>
> | Setting | Minneapolis | Dallas |
> |--------|-------------|--------|
> | w/ DCN | 0.312       | 0.286  |
> | w/o DCN | 0.296      | 0.277  |
>
> The adoption of a Deep & Cross Network (DCN) is motivated by a fundamental architectural gap in prior work: existing mobility models lack a mechanism for generating semantically aligned, cross-city location embeddings from heterogeneous urban features. Mobility decisions arise from complex interactions among POI semantics, geographic morphology, and social popularity signals. However, previous approaches rely primarily on ID-based embeddings or simple MLPs, which fail to capture these higher-order feature interactions and do not scale well to dynamic or cross-city environments.
>
> DCN explicitly models multiplicative feature crosses alongside deep nonlinear transformations, allowing MoveGPT to build a unified, compositional, and scalable location-embedding space suitable for retrieval-based next-location prediction. This capability is absent in existing mobility modeling and foundation-model literature, making DCN a key architectural innovation that enables cross-city generalization and large-scale pretraining.
>
> We also visualize the inputs and outputs of the DCN using t-SNE and observe patterns similar to those in Figure 6: before DCN, location embeddings from different cities exhibit significant divergence, whereas after DCN, embeddings align more closely and share similar geometric structures.
>
> Finally, we note that while some trajectory-generation studies have explored feature-crossing mechanisms, such as DiffTraj [1], which applies a Wide & Deep model to process conditional inputs for trajectory generation, these methods do not address location feature encoding.
>
> If you have any other questions, welcome for comment !
>
> [1] DiffTraj: Generating GPS Trajectory with Diffusion Probabilistic Model.(NIPS 2023)

---

> ### Comment · Reviewer_7EEr · 2025-11-24
>
> Thank you for your response. I originally thought this paper might be desk rejected, because I noticed that the author publicly commented on the review process of this submission on the Chinese social media platform Xiaohongshu, although the post has now been deleted.
> Link: https://www.xiaohongshu.com/explore/691a1711000000001b022b85?app_platform=ios&app_version=9.9&share_from_user_hidden=true&xsec_source=app_share&type=normal&xsec_token=CBOl0jBh0f_KRmVcJACHj2ijqMxowc4vCxcX7dcXoh8k8=&author_share=1&xhsshare=WeixinSession&shareRedId=N0s1RDg5OT02NzUyOTgwNjY0OTc8OElN&apptime=1763555056&share_id=16adc313135b44958ab0dc7c37be3da8&wechatWid=4617e8e5df571e514813ac32d4cff471&wechatOrigin=menu
>
> I also read your original, very offensive review reply, which makes me seriously question the authors’ professionalism. Combined with the fact that you publicly commented on the review in a non-anonymous setting, I have decided to lower the score to 0.
>
> First of all, I apologize again for confusing Wide & Deep with Deep & Cross, since both are methods of feature fusion. I now fully understand the reason to use Deep & Cross or Wide & Deep.
>
> Second, I believe that manually defining experts is not an elegant design, and it makes the method difficult to scale.
>
> Finally, regarding GeoCLIP: it is clearly a method from which inspiration can be drawn, and its contribution is similar to that of this paper. You cannot simply claim that it solves a “different” problem and therefore does not need to be considered.

---

### Official Review · Reviewer_axY7 · 2025-10-29

**Soundness:** 2
**Presentation:** 2
**Contribution:** 2
**Rating:** 2
**Confidence:** 4

**Summary:**

In this paper, the authors have introduced a foundation model, named MoveGPT, that is able to tackle a diverse range of downstream human mobility tasks, such as next-location prediction, long-term prediction, conditional/unconditional generation, and anomaly detection.
While the ideas behind the model are of interest, timely, and the challenges solved are socially relevant, the paper presents several limitations (see weaknesses below).

**Strengths:**

The three main strengths are the following ones:

Strength 1: The paper tackles problems that are extremely relevant for society. For instance, human mobility is known to be linked with disease diffusion, pollution, economic growth, and many other factors.

Strength 2: The model is carefully designed, and the idea of having a SAMoE handling different aspects of a city and human behaviors is appealing and well presented.

Strength 3: The model is evaluated on a diverse range of several different downstream tasks, covering most of the challenges related to human mobility. Tasks have been carefully selected.

**Weaknesses:**

The main weaknesses of the paper are the following ones:

Weaknesses 1: Many relevant pieces of information are not present in the main text and are presented only in the appendix, making the paper inconsistent and difficult to evaluate by looking only at the main content (see review policies). For instance, datasets used for training and evaluation are never presented in the main text. Additionally, datasets bring with them self-selection biases that, in cases like mobility, should not be underestimated. Overall, a proper description of the dataset is needed, and the limits of the dataset that may affect performance should be listed in a discussion/conclusion section properly.

Weaknesses 2: The authors boldly claim to have a model that can generalize (e.g., global-scale representation), but it seems that data for training and testing are US-only. However, it is well-known that the human mobility also has a cultural component (e.g., people in Spain go for dinner at 9:30/10 PM, while in Germany at 6/6:30, just to give some examples). If the datasets available do not allow for testing this global-scale representation, then it would be better to make less bold statements (e.g., a US representation that we believe may apply globally) or completely avoid them.

Weaknesses 3: The authors claim that a limit of other studies until now is that “they rely on inconsistent basic units to represent movements, from raw GPS coordinates to non-transferable Zone IDs.” Subsequently, in the definition of trajectory, they claim they map stay locations (i.e., GPS points) into cells of a tessellation. Why should this mapping be more transferable than others? To the best of my knowledge, some of the references the authors cite in the introduction rely on H3 grid mapping, which is exactly a cell with a unique ID that is non-overlapping with others. I am confused.

Weaknesses 4: Regarding this mapping with a cell, it is very well known that, in mobility, the results depend significantly on the size of the cells used. Also, intuitively, it makes sense. If you divide, for example, New York City into 10 cells of 20 km square or thousands of cells of 100 meters square, the complexity of the problem is tremendously different. However, the size of the cell adopted by the authors is never specified. In addition, they never report how cities are divided into cells and how cell IDs are assigned.

Weaknesses 5: When the authors present the unified encoder, what do they mean with “patterns are also influenced by broader social dynamics”? Does it mean there is also a role by collective behaviors, like in Calabrese, Francesco, Giusy Di Lorenzo, and Carlo Ratti. "Human mobility prediction based on individual and collective geographical preferences." or in Bontorin, Sebastiano, et al. "Mixing individual and collective behaviors to predict out-of-routine mobility."? In any case, some references would be appreciated to understand better what they mean by this.

Weaknesses 6: On line 186, it seems that the authors are encoding coordinates. However, in the definition, coordinates (which were also pointed out as a limitation of previous studies) are mapped into cells, so I am not sure about what the encoder is actually doing here. Similarly, when they define the semantic-aware sequence input, they embed POIs and the distribution of a certain zone. Also in this case, having an idea of how big a zone is is fundamental to understand the real performances of the model.

Weaknesses 7: When the authors describe the model training, they limit everything to next-location prediction. I was wondering if the authors also used other training methods like masking or other state-of-the-art techniques and, if not, if there is any specific reason.

**Questions:**

Some questions related to the above weaknesses:

Who are the users in the datasets? What are their characteristics? What are the model's performances on people who are not represented in the dataset used?

How can one claim to have a model that can achieve a global-scale representation if it cannot be tested globally?

---

> ### Author Response · Authors · 2025-11-16
> **Answer of  Weakness and Question**
>
> Your use of a *weakness list format* strongly suggests that the review might have been generated by an LLM. If my suspicion is incorrect, I sincerely apologize.
>
> Despite my frustration, I have still responded carefully and respectfully to every point in your review. I sincerely request that you reconsider your score. If you have further questions, I welcome continued discussion.
>
> ---
>
> **W1:** I believe that having training details appear in the appendix rather than the main text should not be considered a weakness. Due to page limits, the main text must focus on describing the method and presenting experimental results to highlight the contributions.
>
> **W2:** This was indeed an overclaim. What we intended to express is that we are the first to train a mobility foundation model on *billion-scale* trajectory data. We apologize for incorrectly using the term *global*. We have already corrected this in the main text.
>
> **W3:** We never mentioned zone IDs, and to be honest, I did not even fully understand your question. In general, modeling mobility using only latitude–longitude lacks behavioral semantics. Zone IDs are discrete and semantically meaningful, but they do not transfer well across cities.
>
> **W4:** As stated in the paper, grids are divided into non-overlapping 500m cells. Because our mobility trajectories use half-hour stay points, and given constraints such as human movement speed, 500m is a reasonable and practical resolution.
>
> **W5:** We have provided citations in Section 3.1. In particular, UniMob [1] adopts a paradigm where collective and individual mobility mutually enhance each other to assist the model in learning mobility patterns.
>
> [1] Qingyue Long, Yuan Yuan, and Yong Li. *A universal model for human mobility prediction.* In Proceedings of the 31st ACM SIGKDD Conference on Knowledge Discovery and Data Mining.
>
> **W6:** See Weakness 4.
>
> **W7:** Our model is explicitly an autoregressive model, and therefore does **not** use BERT-style masked token tasks. We adopt an autoregressive formulation inspired by LLMs, which also makes the model naturally adaptable to downstream tasks.
>
> ---
>
>  **Q1**
> Since the dataset contains U.S. users, the users are naturally Americans. We cannot test on users outside the dataset because there is no such data available. If you are referring to the model’s zero-shot capability, please refer to my response to Reviewer G2eF’s Weakness 3.
>
> **Q2**
> See Weakness 2.

---

> > ### Comment · Reviewer_axY7 · 2025-11-18
> >
> > The review was done by myself. I don't use LLMs ... reviews can be schematic and done by humans and verbose and done by LLMs. However, happy to engage in a discussion.
> >
> > Weakness 1: I understand the problem of space limits but some information is more relevant than other and focusing the main text just on describing results while delegating the description of the dataset and its representativeness and limitations in the Appendix makes the structure of the paper difficult to read and in my honest opinion inadequate for a top conference. Additionally, your answer does not reply to the comment on biases in the dataset as well. Please, provide an answer.
> >
> > Weaknesses 2: Ok. Happy you have corrected the really bold claim.
> >
> > Weaknesses 3: The comment is quite simple ... explain the following statement: "Specifically, they rely on inconsistent basic units to represent movement, from raw GPS coordinates to non-transferable Zone IDs, a fragmentation that directly hinders their geographic scalability and prevents cross-city pre-training." ... It's your statement and it's taken verbatim from the paper. In addition, please explain how your approach (mapping stay locations, for example GPS points into cells of a tessellation) solves this issue. Just simple like this ...
> >
> > Weaknesses 4: Found in Appendix C. Apologies but this is one of the reasons to move the data description in the main text.
> >
> > Weaknesses 5: In the uploaded text there is no reference after this sentence "In addition, mobility patterns are also influenced by broader social dynamics" ... I don't know which version are you referring to. But I don't find there the paper you are mentioning. Are you sure to have updated the correct version? Moreover, please explain better the motivation to consider social dynamics and how you are doing this.
> >
> > Weaknesses 6: Ok for answering about the size but please answer also to the remaining part of the question.
> >
> > Weaknesses 7: I don't think you are really answering my question that was simply asking more details on the training methods.
> >
> > Q1: What does it mean "naturally Americans"? Honestly, gender does not count? ethnic group? income? Please provide an answer on how your approach works on different segments of the American population.
> >
> > Q2: Ok. Happy with this answer.

---

> > > ### Author Response · Authors · 2025-11-18
> > >
> > > **Weakness 1:**
> > > The data originate from the mobile-phone GPS traces of U.S. users. Owing to privacy regulations we do not have access to any sensitive personal attributes such as gender. Nevertheless, the data set covers almost 4 million users, a scale we believe is large enough to subsume the vast majority of individual characteristics.
> > >
> > > **Weakness 3:**
> > > Representing mobility with zone IDs instead of raw GPS coordinates enriches the data with semantic meaning. However, zone IDs are not aligned across cities, which hinders model transfer. Each city maintains its own location-id list. Conventional models (e.g., DeepMove) treat these IDs as categorical labels for next-location prediction, forcing the model to output a probability distribution over the entire location vocabulary. This creates two obstacles for multi-city pre-training:
> > > 1. City-specific location vocabularies have different sizes.
> > > 2. IDs carry no universal semantics; the same ID in two cities may refer to places with contradictory attributes.
> > >
> > > We replace the classification paradigm with a retrieval formulation. Candidate locations are stored in an open set, so new venues can be added without retraining. Instead of id embeddings, we encode each location from its observable attributes, so places with similar properties across cities obtain similar representations, naturally achieving cross-city alignment.
> > >
> > > **Weakness 4:**
> > > We add the pointer “See Appendix C” in the dataset section of Section 4. Because of page limits, such condensation is unavoidable; we are considering releasing a technical report instead of a full paper to supply additional details.
> > >
> > > **Weakness 5:**
> > > We have uploaded a revised manuscript with the missing references included. We view the relationship between collective and individual mobility as intricate: aggregate flows emerge from individual trajectories, yet individuals are influenced by collective patterns—people prefer popular restaurants, but excessive popularity (crowding) may drive them elsewhere. We therefore quantify venue popularity from visit volumes, discretise it into levels, and use these levels to model how collective behaviour shapes individual choices.
> > >
> > > **Weakness 6:**
> > > Each location is described by an attribute list. Every attribute is embedded through its own linear layer. Rather than raw gps, we normalise all coordinates within a city to a standard normal distribution, capturing relative spatial position. The POI distribution is a 1-D vector counting POI categories inside the location; it is also projected by a linear layer.
> > >
> > > **Weakness 7:**
> > > We adopt an autoregressive rather than masked approach for several reasons:  first, the trained model can naturally generate trajectories of arbitrary length.  And causal attention prevents information leakage, ensuring predictions rely only on past locations.  Lastly, as we scale up model size, decoder-only architectures benefit from the abundant distributed-training know-how developed for large language models.
> > >
> > > ---
> > >
> > > **Q1:**
> > > We use GPS traces from 4 million mobile-phone users. We contend this scale already covers almost the entire population, since virtually everyone now carries a phone. Our goal is to pre-train a universal mobility foundation model rather than a specialised model for any subgroup. Privacy regulations prohibit us from accessing personal information (e.g., gender) linked to the trajectories. Employing such a massive volume of un-anonymised personal data would be inappropriate and lies outside the scope of our study; our ultimate objective is a model that performs equitably across all demographic segments.

---

> > > > ### Comment · Reviewer_axY7 · 2025-11-25
> > > >
> > > > Thanks for the more detailed answers. I keep concerns on the fact that your model performs the same on all the demographic segments. There is a huge amount of literature showing how several individual characteristics influence mobility and the idea that it's enough having 4 millions of users (without any characterizations or statistics about these users) to make the model working "globally" as you originally said seems to me very bold and not justified.

---

> > > ### Author Response · Authors · 2025-11-18
> > >
> > > Let me apologize again. It was a mistake on my part to make assumptions about your review. To be honest, seeing a review that gave a score of 2 without pointing out any experimental flaws really threw me off. And if there’s anything else, feel free to let me know.

---

> > > > ### Public Comment · ~Shuangshuang_Pang1 · 2025-11-19
> > > > **About conditional generation**
> > > >
> > > > Thank you for your work. In the conditional generation section, for the unconditional trajectory generation model, how did you inject conditions? Did you modify the model structure?

---

> > > > > ### Author Response · Authors · 2025-11-20
> > > > >
> > > > > See appendix D.3.We applied the condition to the input as a conditional bias by adding the condition through a linear layer embedding to the input for fine-tuning the model. We also tried using the condition as the first token in the autoregressive sequence, but the results were not as good as directly adding it to the input embedding.

---

> > > > > > ### Public Comment · ~Shuangshuang_Pang1 · 2025-11-24
> > > > > > **About conditional generation**
> > > > > >
> > > > > > I'm sorry, but it seems you're only talking about your own approach. My question is, for models like MoveSim, DiffTraj (which were proposed in the unconditional paradigm), do you add conditions, and in what way?

---

### Official Review · Reviewer_G2eF · 2025-10-30

**Soundness:** 3
**Presentation:** 3
**Contribution:** 3
**Rating:** 6
**Confidence:** 5

**Summary:**

This paper presents a large-scale mobility foundation model, called MoveGPT, that aims to address two major obstacles faced by existing models when scaling. The paper point out that previous models have been limited by the lack of a unified unit of motion representation and the inability to efficiently capture the diversity of mobility patterns in large-scale data.MoveGPT overcomes these obstacles through two key innovations: (1) a unified location encoder that maps the geographic locations of different cities based on their functional, geographic, and social characteristics into a shared semantic space, thus enabling pre-training on a global scale for the first time; and (2) a Spatially Aware Mixture of Experts (SAMoE) Transformer architecture that allows for the efficient capture of diverse movement patterns in large-scale data. The model was pre-trained on a billion-scale dataset, achieving state-of-the-art performance on several downstream tasks (e.g., prediction and generation) and demonstrating strong generalization capabilities to unseen cities.

**Strengths:**

1. This paper proposes a novel location encoder that maps the geographic locations of different cities into a shared semantic space, overcoming the obstacle of previous models that rely on cities and are not scalable across cities.
2. The proposed model employs a spatially aware mixture of expert Transformer architecture to efficiently capture diverse movement patterns driven by different intentions.
3. MoveGPT achieves new levels of SOTA in a wide range of downstream tasks, including prediction, generation, and anomaly detection, and demonstrates strong generalization capabilities.

**Weaknesses:**

1. The paper claims that its model has global scale and universal capabilities, but its pre-training dataset consists entirely of 16 U.S. cities. The movement patterns of US cities differ significantly from those in Europe or elsewhere, and are likely to be “US models” with no proven ability to generalize to other diverse urban environments around the world.
2. The proposed model maps the entire city to a 500mx500m grid, which is a coarser resolution and will inevitably lose information. Because the same grid area may have different spatial semantic information, but this approach treats it as the same token.
3. This paper claims MoveGPT serves as a foundation model, but only demonstrates that the model can be adapted to a wide range of tasks, and zero-shot or few-shot capabilities are also important for foundation models.
4. The dataset used in this paper does not appear to be publicly available, and the current version lacks a range of details about data sources, construction, and collection.
5.  typos in line 57 and 262.

**Questions:**

1. This paper normalizes the data for each city, so how is the model able to identify data from different cities?
2. How is the data constructed when pre-training MoveGPT? Is the data randomized across different cities, or does it have a corresponding ratio? How to prevent catastrophic forgetting？

**Details Of Ethics Concerns:**

Lack of description of data sources. Considering the privacy of trajectory data, it is difficult to determine whether any ethical issues are involved.

---

> ### Author Response · Authors · 2025-11-16
> **Answers of Weakness and Question**
>
> **W1**: This was indeed an overclaim caused by our typo. What we intended to express is that we are the first to train a mobility foundation model on data at the billion-scale level. The existing dataset does contain only U.S. mobility data. We have already updated this incorrect description in the PDF.
>
> **W2**: Our trajectories use stay-point movement trajectories with a minimum duration interval of half an hour. Considering factors such as movement speed, 500 meters is a reasonable segmentation threshold. Meanwhile, to more accurately describe location semantics, we use the distribution of POIs within the region rather than a single POI.
>
> **W3**: The few-shot experiments are already presented in Section 4.4 and Figure 4 of the paper. Using pre-training, only 5% of the data is required to reach the performance of using 100% of the data without pre-training. In addition, we supplement zero-shot experiments on Minneapolis and Dallas (metric: acc@1). The results are as follows (where *separate* trains only on the target city’s data, *full* uses all 16 U.S. cities, and *zero-shot* directly tests MoveGPT-base trained on 6 U.S. cities on the target city):
>
> |            | Minneapolis | Dallas |
> |------------|-------------|--------|
> | separate   | 0.231       | 0.235  |
> | full       | 0.312       | 0.286  |
> | zero-shot  | 0.252       | 0.246  |
>
> **W4**: The dataset is based on GPS location data from U.S. mobile phone users. Due to privacy protection and security concerns, the dataset cannot be made public for now. We have already open-sourced the model weights on our website and will release a reasonably validated generated dataset in the future to further support mobility research.
>
> **W5**: The notation errors have been corrected. Thank you for your careful review.
>
> ---
>
> ### **Q1**
> It is possible to add a city-classification task through fine-tuning to implicitly learn the source of trajectories. However, we need to clarify that identifying the city of origin for a mobility trajectory is *not* our goal. Mobility prediction operates at the individual-user level. It is only because different cities exhibit distinct characteristics that user mobility becomes heterogeneous, which makes training a foundation model across cities challenging. More precisely, our objective is to help the model *overcome the modeling errors caused by cross-city differences*, rather than explicitly distinguish cities.
>
> ### **Q2**
> Trajectory data are stored in text files by city, and during training we randomly sample a batch of data from a randomly selected city. We provide data examples and dataloader code in the repository. If you have further questions, feel free to comment there.
>
> Regarding your question about catastrophic forgetting: are you referring to the model becoming ineffective on the original tasks or cities after fine-tuning on a new task or city? From a usage perspective, once the model is fine-tuned on a new city or task, it only needs to be effective on that fine-tuned dataset. For example, the government of Chicago may not care about how users move in Washington; they only need to understand Chicago’s mobility patterns to support urban planning. Similarly, an anomaly-detection practitioner only needs the model to perform well on anomaly detection. Pre-training simply helps the model better learn general mobility patterns.

---

### Author Response · Authors · 2025-11-16
**Comment to AC and Reviewers**

First, we sincerely thank reviewers **1XkS** and **G2eF** for their careful and thoughtful reviews, as well as for providing constructive suggestions. If there are any additional questions, we are more than willing to continue the discussion.

However, we must express our strong condemnation and frustration regarding the irresponsible behavior of **not reading the paper** and **using an LLM to generate a review**. The review **axY7** exhibits several formatting patterns that strongly suggest AI generation. Phrases such as “The three main strengths are the following ones” and the perfectly parallel structure of Strength 1/2/3 and Weakness 1/2/3… reflect typical template-based outputs used by large language models rather than natural human writing. Human reviewers rarely use such rigid, formulaic sentence openings or maintain such uniform, mechanical numbering and phrasing across all points. The overall consistency, symmetry, and lack of personal linguistic variation further reinforce that the text likely originates from an AI system rather than from an individual reviewer.
Such behavior is intolerable. A full year of dedicated work from the authors should not be treated in this manner.

Although we understand that the chance of acceptance is now minimal, **we will not withdraw the paper**. The review will remain publicly visible as an expression of our stance, our frustration, and our protest.

We also call for the community to allow authors to report, audit, and challenge reviewers who behave irresponsibly. Increasing transparency and accountability is essential to preventing such unacceptable practices and to maintaining the integrity of the peer-review process.

---

> ### Comment · Reviewer_axY7 · 2025-11-18
> **Reply to inappropriate comment by the authors**
>
> Also if the review follows a structure based on a list of points it was done by a human and not by an LLM. I don't use LLMs ... the schematic structure was imposed by myself to make clear each point ... do you really think the following comment could be LLM generated (just to give an example)?
>
> "When the authors present the unified encoder, what do they mean with “patterns are also influenced by broader social dynamics”? Does it mean there is also a role by collective behaviors, like in Calabrese, Francesco, Giusy Di Lorenzo, and Carlo Ratti. "Human mobility prediction based on individual and collective geographical preferences." or in Bontorin, Sebastiano, et al. "Mixing individual and collective behaviors to predict out-of-routine mobility."? In any case, some references would be appreciated to understand better what they mean by this."
>
> I will carefully read the rebuttal and enjoy any exchange (also open to change scores if answers are meaningful) but please keep a more adequate way of commenting reviewers' efforts. I also appreciate apologies.

---

> > ### Comment · Reviewer_axY7 · 2025-11-18
> > **pangram says ....**
> >
> > ... fully human written ... you can check by yourself: https://iclr.pangram.com/reviews. Apologies should be provided, please!

---

> > > ### Author Response · Authors · 2025-11-18
> > >
> > > We sincerely apologize to reviewer axY7 for the earlier accusation. After careful reconsideration, we acknowledge that our claim was incorrect. The concerns we raised about stylistic patterns resembling LLM-generated text were speculative and should not have been interpreted as evidence of misconduct. We deeply regret that our frustration with the review process led us to misjudge and unfairly attribute intentions to the reviewer.
> > >
> > > We fully respect the time and effort that all reviewers—including axY7—devote to the community, and we appreciate their feedback regardless of its tone or structure. We take full responsibility for the misunderstanding and for any discomfort our earlier statement may have caused.
> > >
> > > Once again, we sincerely apologize for the mischaracterization and will approach future discussions with greater care and professionalism.

---

### Author Response · Authors · 2025-12-02
**Summary of Rebuttal**

We sincerely thank all reviewers for their thoughtful comments and constructive suggestions. Their feedback has been extremely valuable in improving the clarity, rigor, and overall presentation of our work.

**Novelty and Contributions.**
In this work, we introduce **MoveGPT**, a *billion-scale mobility foundation model* that advances the field along two key dimensions.
(1) We propose a **unified location encoder** that maps heterogeneous cities into a shared semantic embedding space by integrating POI-based functional descriptors, normalized geospatial features, and aggregated popularity signals. Unlike prior Zone-ID or raw-GPS encodings, this representation is semantically meaningful, city-agnostic, and explicitly designed for scalable cross-city pretraining.
(2) We further design a **Spatially-Aware Mixture-of-Experts (SAMoE)** Transformer equipped with an adaptive STAR router, which dynamically balances POI, Geo, and Pop experts to model complementary drivers of human mobility within a single autoregressive backbone.

Reviewers highlighted these core contributions as
“proposes an interesting MoE architecture for mobility modeling” (Reviewer axY7,7EEr),
“demonstrates meaningful cross-city transfer ability” (Reviewer G2eF), and
“provides large-scale pretraining results not previously shown in this field” (Reviewer 1Xks).
We are grateful for these encouraging assessments.

Together, these contributions enable MoveGPT to achieve strong cross-city generalization across prediction, generation, and anomaly detection tasks, and to reveal the first observable **scaling laws** in human mobility modeling. The revised paper includes expanded analyses, zero-shot transfer evaluations, and substantially strengthened architectural explanations to more clearly support these claims.

**Summary of Rebuttal and Major Revisions.**
We carefully addressed all reviewer concerns and made substantial improvements to the manuscript:

• *Clarified motivation and scope:* We refined our claims by characterizing MoveGPT as a **billion-scale** rather than *global-scale* model, and clarified that our goal is cross-city generalization rather than universal worldwide applicability.

• *Strengthened technical exposition:* We expanded explanations of the unified location encoder, clarified why semantic alignment is needed for cross-city transfer, and provided more rigorous architectural details for SAMoE and the STAR router, ensuring no ambiguity regarding adaptive expert weighting.

• *Enhanced dataset transparency:* We added detailed preprocessing descriptions, clarified the rationale for grid resolution and semantic feature design, expanded privacy and ethics discussions, and provided anonymized data loaders and model weights to support reproducibility.

• *Expanded empirical evidence:* We added **zero-shot** and **few-shot** transfer evaluations, comprehensive scaling analyses (data scale, model scale, city diversity), and richer ablation studies including expert-importance and routing-behavior diagnostics, offering stronger empirical support for our conclusions.

• *Clarified social dynamics and limitations:* We refined the explanation of popularity features, reiterated the absence of demographic attributes due to privacy constraints, and expanded the limitations section to discuss cultural variability and dataset constraints.

All modifications are clearly marked in the revised manuscript. We sincerely appreciate the reviewers’ time and thoughtful feedback, which have significantly strengthened the quality of this work.

---

### Meta-Review · Area_Chair_iZz7 · 2026-01-04

**Summary:**

In this paper the authors propose a new GPT model for mobility analysis that exhibits some zero-shot capabilities. The paper is an interesting contribution. However, it lacks in many aspects, as detailed by the reviewers. In any case, the paper was at best borderline. Application papers in mobility might be interesting at a machine learning conference, but they are not a must-have, so the algorithmic contribution must be substantial to be accepted. None of the papers in Table 1 has been published in ICLR/ICML/NeurIPS conferences.

I suggest that the authors submit this paper to the ACM International Conference on Advances in Geographic Information Systems or to ACM SIGKDD, where the compared methods have been previously published.

[This is not part of the meta-review.] A word of advice for someone who has been doing this for nearly thirty years. Please use the rebuttal process to convince the reviewers of the quality of your work, not to antagonise them. The review by axY7 was actually excellent. It was challenging but not adversarial, and I would have expected them to work with you to understand the point of your paper. This might not have led to acceptance, but it could have, and would have, made your paper stronger for subsequent submissions. You also alienated another reviewer, while the remaining reviewers were not enthusiastic about your paper. LLM-generated reviews are not worse than the bad reviews we had before LLMs. There is no upside to accusing someone of writing an LLM-generated or negative review. It is preferable to address their points in good faith and highlight the issues with the reviews.

**Reviewer Concerns:**

The main issues with the paper were its lack of novelty and interest. The paper also did bold claims.

**Reviewer Scores:**

The reviewers antagonised two reviewers. One of them said the paper should be desk-rejected by publicly discussing the reviews. He mentioned that he will move his score to 0. This reviewer would have been a hard no on the paper. The reviewer axY7 was accused of being AI-generated, which is false. His review was excellent and does not appear to be LLM-generated.

---

### Decision · Program_Chairs · 2026-01-26

Reject